# A novel multi-word paradigm for investigating semantic context effects in language production

**Cornelia van Scherpenberg**[1,2,3]*, **Rasha Abdel Rahman**[1,4], **Hellmuth Obrig**[1,2,3]

**1** Berlin School of Mind and Brain, Humboldt-Universität zu Berlin, Berlin, Germany, **2** Department of Neurology, Max Planck Institute for Human Cognitive and Brain Sciences, Leipzig, Germany, **3** Clinic for Cognitive Neurology, University Clinic Leipzig, Leipzig, Germany, **4** Department of Neurocognitive Psychology, Humboldt-Universität zu Berlin, Berlin, Germany

* cornelia.vanscherpenberg@hu-berlin.de

**Data Availability Statement:** Data and analyses scripts are available at https://osf.io/jq7vu/.

**Funding:** During the preparation of this paper, CvS was funded by the Berlin School of Mind and Brain and the German National Academic Foundation.

## Abstract

Semantic context modulates precision and speed of language production. Using different experimental designs including the Picture-Word-Interference (PWI) paradigm, it has consistently been shown that categorically related distractor words (e.g., cat) inhibit retrieval of the target picture name (dog). Here we introduce a novel variant of the PWI paradigm in which we present 8 words prior to a to be named target picture. Within this set, the number of words categorically related was varied between 3 and 5, and the picture to be named was either related or unrelated to the respective category. To disentangle interacting effects of semantic context we combined different naming paradigms manipulating the number of competitors and assessing the effect of repeated naming instances. Evaluating processing of the cohort by eye-tracking provided us with a metric of the (implicit) recognition of the semantic cohort. Results replicate the interference effect in that overall naming of pictures categorically related to the distractor set was slower compared to unrelated pictures. However, interference did not increase with increasing number of distractors. Tracking this effect across naming repetitions, we found that interference is prominent at the first naming instance of every picture only, whereby it is stable across distractor conditions, but dissipates across the experiment. Regarding eye-tracking our data show that participants fixated longer on semantically related items, indicating the identification of the lexico-semantic cohort. Our findings confirm the validity of the novel paradigm and indicate that besides interference during first exposure, repeated exposure to the semantic context may facilitate picture naming and counteract lexical interference.

## Introduction

The way speakers select appropriate words in a given context has been the subject of research for many decades. It has been shown that both linguistic and task-related factors play key roles in determining which word a healthy speaker will select during language production. Models

The funders had no role in study design, data collection and analysis, decision to publish, or preparation of the manuscript.

**Competing interests:** The authors have declared that no competing interests exist.

to describe uncompromised language production mostly agree in assuming four steps in the word retrieval process most commonly investigated by picture naming [e.g., 1,2]: [1] (Visual) object identification, [2] access to an object's semantic representation, [3] retrieval of the corresponding lexical representation and [4] retrieval of the phonological word form. Notably, the semantic context of the target to be named has been found to influence speed and accuracy of target word production. To explore this finding further the first aim of the current study was to investigate whether naming speed can be modulated by changing the intensity of semantic context activation. This was achieved by modulating the number of items creating the semantic context. The second aim was to assess how speakers explore the visually presented semantic context and whether processing intensity influences naming latencies.

Semantic context effects on target word production have been shown using a number of variations of three classical paradigms: Picture Word Interference (PWI) [distractor word competing with picture; e.g., 3–7], blocked-cyclic naming [semantically homogeneous/heterogeneous blocks; e.g., 8–11] and continuous naming [semantically related interspersed with unrelated items, 12]. The converging observation is that semantic context can influence lexical-semantic processing and lexical retrieval in opposite directions (i.e. both interference and facilitation). This has led to different theories explaining how semantic context interacts with the target, one major debate being at which steps during word production it does so. The "Swinging Lexical Network" (SLN) account by Abdel Rahman et al. [13,14] agrees with many other theories that lexical selection for word production is characterized by competition between lexical entries. In addition, it assumes that a distractor primes the target on the conceptual level, because both share semantic features (e.g., *cat*, *cow*, *pig*, all share the meaning "animal with four legs"). The trade-off between this conceptual facilitation and lexical competition determines whether lexical selection will be inhibited or facilitated. Abdel Rahman et al. [13] argue for a selection mechanism like the Luce ratio [15]. The selection of a target lemma is dependent on the sum activation of all other lemmas. Consequently, the number of activated items in the lexical network and their activation levels should influence the probability of target lemma selection. When many competitors are activated, the target stands in a one-to-many competition with them. The SLN model therefore predicts that only when a cohort of inter-related items induces overall activation in the lexical network, this will surpass conceptual facilitation, and an interference effect will arise. Additional members of the lexical cohort should therefore lead to more activation within the network, and increase interference with the target word [13,14].

So far, this mechanism has been studied mostly indirectly by manipulating the proximity of semantically related items within the naming context. For example, a study by Rabovsky et al. [16] showed that an object is more likely to co-activate mutually related concepts and their lexical representations, the more semantic features it shares with other concepts [17]. Here, pictures with higher endogenous semantic neighborhood densities were named more slowly and less accurately, because they activated a larger cohort of lexical competitors resulting in slower lexical selection.

Moreover, the semantic context paradigms mentioned above have shown that the activation strength of competing items is another important factor. For instance, closely semantically related items that share more semantic features (e.g., *donkey*, *horse*, *cow* vs. *donkey*, *trout*, *owl*), lead to slower naming than semantically distant items. These graded semantic effects have been found for all major paradigms: PWI [18,19], blocked cyclic naming [20], and continuous naming [21]. The findings reveal that semantic interference can be modulated by changing the structure of the semantic context in which a picture is named. One extreme case is that facilitation as opposed to interference is elicited, usually when the semantic relationship between target and (distractor) context is not categorical but associative (e.g. *donkey—stable*, *hay*, *farmer*).

In the framework of the SLN model the explanation is that no interrelated lexical cohort is activated and target and distractor (simultaneously presented or previously named) stand in a one-to-one competitive relationship with each other. In this case the facilitation on the conceptual activation level outweighs interference, and target selection is faster [13,22,23]. Alternative explanations have claimed semantic facilitation to be the default effect, with semantic interference occurring only at post-lexical processing steps, where task-relevant (i.e. semantically related) responses to pictures have to be actively excluded [response exclusion hypothesis; 24–26].

In the present study we focus on categorical semantic relations and investigate whether manipulating the extent of lexical activation within a lexical cohort modulates inhibition on subsequent picture naming. Using a set of closely related entries of a number of lexical cohorts the activation strength per item can be assumed to be largely homogeneous. Using these sets we parametrically change the number of distractors to investigate, whether this has a direct influence on the amount of semantic interference. In this vein a previous study [27] found a significantly increased interference effect in the PWI paradigm when two instead of one semantically related words were shown as distractors. In the present study the semantic context is created by presenting a total of 8 words. Critically, three to five of these words are categorically related, forming the lexical cohort. We measure the influence of cohort size on reaction times when naming a picture presented *after* the word array. The picture to be named is either categorically related or unrelated to the lexical cohort. We hypothesize that reaction times for a related picture will be slower the more related words were presented, because a more strongly activated lexical cohort should lead to more competition between lexical entries, resulting in longer naming latencies.

The extent to which presenting a number of written words before naming pictures can influence picture naming speed has been investigated in previous research [28–31]. However, in these experiments, words were presented consecutively and had to be overtly read out aloud. Moreover, the findings are partially contradictory. For example, Navarrete et al. [28 (Experiment 3), 29] found no transfer of interference from word to picture naming within one semantic category, whereas Vitkovitch et al. [30,31] did report semantic interference for naming pictures after having named semantically related words. We here investigate how simultaneous presentation and lexical activation by reading (not producing) the words impact on the processing of the semantic relationships between the words and consecutive naming of un/related items.

To study and control how participants process the semantic context we additionally measure their eye movements while they view the distractor words. We proceed from the rationale that eye tracking can be used to investigate the semantic 'competence' of viewers. This assumption rests on paradigms performed in people with Primary Progressive Aphasia (PPA) and neurotypical controls. Suggesting impaired semantic memory abilities, participants suffering from PPA [32,33] fixated on semantically unrelated objects (foils) more often and longer when compared to neurotypical controls, likely indicating difficulties to establish the semantic relationships between concepts. Here we hypothesize that the neurotypical young adults are semantically competent and should hence fixate on words longer which they have recognized to belong to the same category, when compared to the unrelated words. Thus, analysis of fixation times was used to investigate the semantic 'competence' of the participants in each trial. Additionally, we can use this measure to estimate the extent to which they activate the lexical cohort. According to the eye-mind hypothesis [34,35], readers' gaze durations are immediately linked to what they are processing. That is, words that are fixated longer are also processed longer. We therefore predict that the longer participants fixate on related words belonging to the lexical cohort, the more activity will spread to this cohort and induce stronger competition resulting in inhibition on target selection.

Apart from the nature and extent of semantic relation it should be noted that previous research has shown interference and facilitation to differ as a function of timing (at the trial level) and repetition (i.e. across the experiment).

*Timing* in the PWI paradigm has been shown to greatly affect naming speed: Prominently, the interval between a distractor word and target (the stimulus onset asynchrony, SOA) influences the polarity of the context effect [5,36–38]. Manipulating the SOA systematically with different time intervals, Zhang et al. [36] demonstrated that a semantic interference effect from categorically related word distractors only occurred at an SOA of -100ms before, or of 0ms, that is simultaneously to, target onset. At longer negative SOAs (-1000 to -400ms), the effect transformed into semantic facilitation–using the same stimulus materials. Similarly Python et al. [38] find facilitation from categorically and associatively related distractor words at an SOA of -400ms. These findings indicate that at longer SOAs, conceptual priming outweighs lexical competition. We will address this issue in more detail in the Discussion. Moreover, semantic context effects may change when a specific picture or a category is *repeatedly* named. For example, in the blocked cyclic naming paradigm's first presentation cycle, a homogeneous block often does not lead to longer but shorter naming latencies when compared to the first heterogeneous block [9,22,39–42]. Interference from homogenous context appears only from the second cycle onwards, and has been reported to grow with each repeated block of related pictures [growth effect; e.g., 9,11; but see 8, and 39 (Experiment 1 and 2a)]. In continuous naming, reaction times increase across ordinal position of the target pictures within their semantic category [e.g., 8,12,28,43]. These cumulative or growth effects are explained by incremental learning as proposed by Becker et al. [44] and Damian and Als [9] and further developed in a computational model (the "Dark Side Model") by Oppenheim et al. [45]. It is assumed that connections between a concept's features and its lexical representation are strengthened by repeated access during target naming. This results in faster activation of the item and therefore reduced naming latencies on future naming occasions (repetition priming). However, enhanced activation makes the already named item a stronger competitor for its related concepts, while connections to semantic features shared between the target and related concepts from the same semantic category are weakened (the "dark side" of repetition priming). Therefore, access to a related concept's lexical representation is slower. Conceivably a combination of both factors leads to cumulative interference for items from one semantic category in picture naming settings such as the continuous or blocked naming paradigms [28,45,46].

In contrast to these paradigms, to our knowledge, for a PWI paradigm changes across naming repetitions have been formally addressed only in one recent study [47]. Using an auditory PWI design, interference effects are reported to be largely stable across naming repetitions of the same pictures with phonological distractors. This stands in contrast to the other paradigms mentioned above, and systematic conclusions about the stability of the interference effect in PWI paradigms can only be tentative at present.

The repetition- or sequence-effect changing the contribution of interference and facilitation across the experiment is complemented by findings of studies looking at small-scale changes of the effects in response time distributions. Two recent studies have shown that when dividing the participants' rank-ordered response times into deciles, the interference effect is driven by the slowest decile and small or absent in the fastest 10% of response times [48,49]. Both studies explain findings by attentional processes which influence the strength of distractor processing: When attention is low, the distractor might be processed more intensely while the ability to inhibit its interfering effect might be reduced, and therefore reaction times are longer. A high level of attention, however, mediates the interference effect and reaction times become faster.

All in all, research on the change of interference and facilitation effects as a function of tim-ing (SOA) manipulations, over repeated naming-instances and within response time distribu-tions, shows that the effects are sensitive to timing modulations and can sometimes even occur in one and the same task. The present study therefore addresses this issue by including the rep-etition factor in the analyses. We aim to explore whether the typical interference effect–repli-cated many times for the PWI paradigm–can be influenced by trial progression as well. Repeated access to the same category members might facilitate target retrieval across several naming occasions. Alternatively, it might lead to increased competition within the category's lexical cohort and therefore to cumulative interference as the experiment progresses. This pro-cess may be influenced by changes in attention across trials. Finally, a long SOA, necessary to allow for full processing of each of the eight distractor words, might affect the semantic inter-ference effect as well.

## Methods

### Participants

24 young adults (15 females), aged 18–32 years (M = 24.5, SD = 3.8), participated in this study in return for monetary compensation of €9 per hour. All participants were right-handed, had no history of neurological or other relevant diseases and had normal or corrected-to-normal vision. The number of participants was determined through the randomization lists needed to fully randomize all stimuli and trial orders (see below).

Experimental procedures were approved by the Institutional Review Board of the Univer-sity of Leipzig, Germany, in accordance with the Declaration of Helsinki, and written informed consent was obtained from all participants.

### Material

We used a variation of the picture-word interference (PWI) approach in which a picture has to be named after the presentation of a distractor word. Different form 'classical' PWI-designs, an array of 8 distractor words was presented simultaneously, before the picture to be named appeared. Thereby the number of related and unrelated distractor words could be parametri-cally varied.

The stimulus set consisted of 42 items from 7 semantic categories. The chosen items were closely related as members of subcategories of superordinate categories (*superordinate catego-ries* in brackets): seating (*furniture*), street-vehicles (*vehicles*), face parts (*body-parts*), fruits (*food*), upper body clothing (*clothes*), hoofed animals (*animals*), and carpenter's tools (*tools*); see S1 Appendix).

The frequency of occurrence as a target picture to be named and as a member of the distrac-tor word set was equal across items. Within the sets of eight words a varying number [3, 4 or 5] belonged to the same semantic category, representing the lexical cohort. The remaining unrelated items [i.e. 5, 4 or 3] each stemmed from one of the remaining semantic categories. To control for potential confounding effects all words used in the paradigm have a highly simi-lar frequency: mean = 12.29, sd = 1.88, according to the Leipzig Corpora Collection [50]. Moreover, potential item-based effects are strongly attenuated by the fact that randomization was complete across conditions: Each target picture was named once as a related or unrelated target in each of the three distractor conditions, that is: following the presentation of three, four or five related words within the lexical cohort, whereby the cohort was always randomly arranged from one of the 7 categories. With the 7 semantic categories with 6 items each and each picture being named 6 times in total, this led to a total number of 252 trials. Out of these, 84 trials each were attributed to one experimental block.

The word stimuli were presented in white Arial font, size 40, on a black screen. All pictures were colored photographs taken form the Bank of Standardized Stimuli [51], stock image databases or creative commons sources. They were scaled to 5.8 x 5.8 cm (300x300 pixels, 5.5˚ of visual angle at a distance of 60cm between the viewer's eyes and the screen). The material was selected avoiding strong visual similarities between members of small categories, e.g. "apple" and "grapes" for fruits. A complete list of the stimuli is given in the supplementary materials.

## Apparatus

The stimuli were presented using the Psychophysics Toolbox extension [52] for MATLAB (2017a, MathWorks, Inc.) on a Lenovo ThinkPad T420 laptop (14" monitor, 1600x900 pixels resolution). Eye movements were recorded from both eyes using a Tobii X2-60 eye tracker with a 60 Hertz sampling rate. Voice responses were recorded using a Blue Yeti USB microphone.

## Design and procedure

The variation of the number of related words in the distractor set results in a 2x3 design with picture TYPE (related vs unrelated) and SIZE of lexical cohort [3, 4 or 5] as within-participants factors. Twelve randomized lists were created with the constraints that target pictures were separated by a minimum of two other items and that each target appeared once with a related and once as an unrelated distractor set in each block. Across each list, the participants therefore named each item six times. The lists were duplicated and randomly assigned to the 24 participants.

At the start of each session participants were instructed about the experimental procedure and were then seated in a dimly lit, sound-proof room in front of the laptop and eye tracker with a distance of approximately 60 cm to the screen. A chin rest was used to minimize head movements and improve eye-tracking data quality.

Prior to the main experiment participants were familiarized with the pictures: each picture was presented once with the written name centered on a black screen, which participants read out aloud. The familiarization phase was self-paced and the order of picture presentation within this phase was randomized individually for each participant. No participant had difficulty recognizing and naming the pictures. After familiarization the eye tracker was calibrated according to a 5-point calibration procedure. This was followed by three practice trials, after which any remaining questions were addressed by the experimenter.

The experimental sessions consisted of three blocks with 84 trials each. Between blocks, participants were able to take a break. Each trial started with a fixation cross centered on a black screen (0.5s), directly followed by a set of eight words presented in a circle around the center of the screen for 6s (see Fig 1 for a typical trial procedure). Participants were told that a minimum of three of the eight words were related to each other and they were instructed to inspect the word set freely. During the viewing part, participants' eye movements were recorded by a Tobii X2-60 eye tracker. Directly after, the distractor words disappeared, and the target picture was presented for 2s. Participants were instructed to name the picture as quickly and accurately as possible. After an inter-trial interval of 0.5s, the next trial started automatically. Each trial lasted for 9s, resulting in a total experiment time of around 38 minutes, not including breaks.

## Analysis

### Reaction times

The voice onset times were detected using Chronset [53], and checked manually using Praat [54]. The onsets were determined at the start of each word, excluding stuttering or "uhms". 3.14% of all trials had to be excluded from further analyses. 2.36% were trials in which

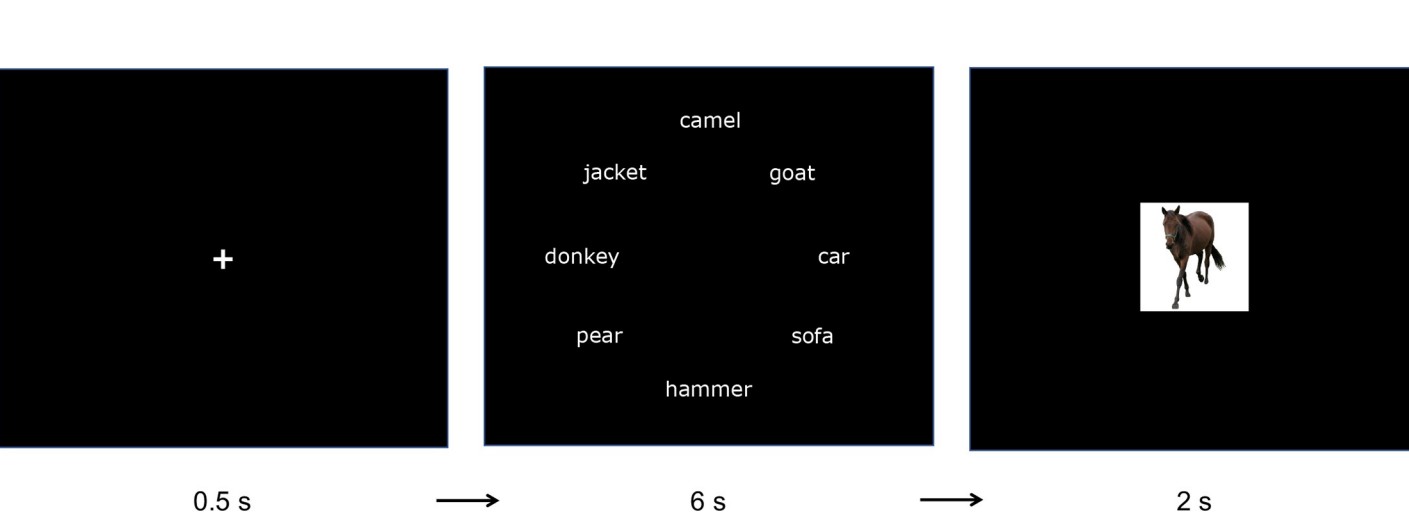

**Fig 1. Exemplary procedure of a trial in which the word set contains a lexical cohort of three items from the semantic category "hoofed animals" and this lexical cohort is related to the target picture.** In the actual experiment, the words were presented in German.

participants did not respond at all or the recording was cut off, whereas only 0.78% were due to false responses. Errors were therefore not analyzed any further.

## Eye tracking data

From the raw data samples fixations and saccades were detected using the GazePath algorithm [55] on the mean x- and y-coordinates of the left and right eye. Heatmaps of the fixations were plotted to establish large enough but not overlapping Areas of Interest (AoI) for each word in the circular word set. These were then defined as rectangles of 270x170 pixels around each word. Trials where GazePath had failed to detect any fixations were excluded from analysis. This led to a total data loss of 1.87% for the eye tracking data. Combining data loss from reaction time and eye tracking data, a total of 5% had to be removed from data analysis.

## Statistical analysis

All statistical analyses were performed using R version 3.6.1 [56]. Generalized Linear mixed effect models (GLMM) were run with random slopes for subjects and items, using the lme4 package in R for linear mixed models [version 1.1–21;, 57], and $p$ values were determined using the package lmerTest [58]. This allowed us to investigate the relationship between voice onset times and picture type, number of related items in the word set, and fixation durations on related items for the group, while taking individual and stimulus-related variance into account. We always started with a model including the maximal random structure. When convergence errors occurred, we reduced the model by running principal component analyses on the random-effects variance-covariance estimates and correlation parameters until the random structure was supported and convergence achieved [59–61]. As suggested by Lo et al. [62], reaction time data can be best modelled using GLMMs to approximate normal distribution of the data without the need to transform the raw data using inverse or log transformations. For the present analyses we chose a Gamma distribution with identity link, to best match the right-skewedness of the raw data with a long tail in the slow RTs, and also in the fixation durations distribution (see S2 Appendix).

**Table 1. Mean RTs in milliseconds and standard error of the means for each naming condition.**

| Distractor set size | 3 | | 4 | | 5 | | total | |
|---|---|---|---|---|---|---|---|---|
| Picture type | related | unrelated | related | unrelated | related | unrelated | related | unrelated |
| Mean RTs in ms | 847.06 | 833.13 | 829.82 | 827.82 | 835.99 | 821.41 | 837.69 | 827.44 |
| SE | 6.05 | 6.15 | 6.13 | 6.03 | 6.17 | 5.86 | 4.56 | 4.48 |
| Interference | 13.93 | | 2.00 | | 14.58 | | 10.25 | |

SEM = Standard Error of the Mean. Values are adjusted for within-participant designs following [63].

## Results

### Reaction times

Raw naming latencies for picture type in total and in each distractor set condition are given in Table 1.

To statistically confirm the differences in naming latencies between picture types (related or unrelated to the distractor set), distractor set sizes [3,4 or 5 related words], and naming repetitions, we used generalized linear mixed models (GLMM). We report estimates, standard errors, t- and p-values in the text and tables for complex models. All full models and model outcomes can be found in S2 Appendix (Tables B1 and B2).

**Relationship between picture TYPE and distractor set SIZE.** We first turn to the analysis of the global effects on naming latencies, that is the main effect of picture TYPE, the main effect of distractor set SIZE, as well as the interaction between the two. In this first model, picture type and set size were both contrast-coded using sliding difference contrasts, which compute differences between adjacent factor levels. This allows to retrieve pairwise comparisons directly from the model output, instead of running post-hoc analyses (e.g., related vs unrelated picture type, 4 vs 3 set size; note, however, that we can only compare n-1 factor levels in each model). The final model that converged included a fully specified random structure (by-subject and by-item random intercepts and random slopes for all fixed effects plus interactions), excluding correlation parameters. It revealed a significant semantic interference effect, in that naming a related picture was slower than naming an unrelated picture (TYPE; estimate = 10.94, *SE* = 4.00, *t* = 2.73, *p* = 0.006). The main effect of set SIZE was significant for 4 compared to 3 distractor words (estimate = -11.33, *SE* = 4.13, *t* = -2.74, *p* = 0.006) and for 5 compared to 3 distractor words (estimate = -11.79, *SE* = 4.05, *t* = -2.91, *p* = 0.004). This indicates that naming was significantly faster for 5 or 4 distractor words compared to only 3 distractor words. The interaction between picture TYPE and set SIZE was significant for 4 vs 3 distractor words (estimate = -11.98, *SE* = 5.21, *t* = -2.3, *p* = 0.021) but not for 5 vs 3 words (estimate = 0.46, *SE* = 5.87, *t* = 0.08, *p* = 0.938). These main effects are summarized in Fig 2.

To investigate this interaction further we fitted another model, where the fixed effect of picture type was nested within the levels of distractor set size [64]. The random structure was again fully specified, without correlation parameters. The results show that interference was only significant at a set size of 3 (estimate = 14.01, *SE* = 5.11, *t* = 2.71, *p* = 0.006) and 5 (estimate = 15.63, *SE* = 5.51, *t* = 2.83, *p* = 0.005) but not 4 distractor words (estimate = 2.75, *SE* = 5.02, *t* = 0.55, *p* = 0.584), in line with the interaction effects in the first model. These results show that contrary to our hypothesis, interference did not increase for additional distractor words.

**Relationship between picture TYPE, distractor set SIZE, and naming REPETITION.** We furthermore fitted a GLMM to track the development of the interference effect and the effect of set size across naming repetitions. Here picture repetition was added as a continuous fixed

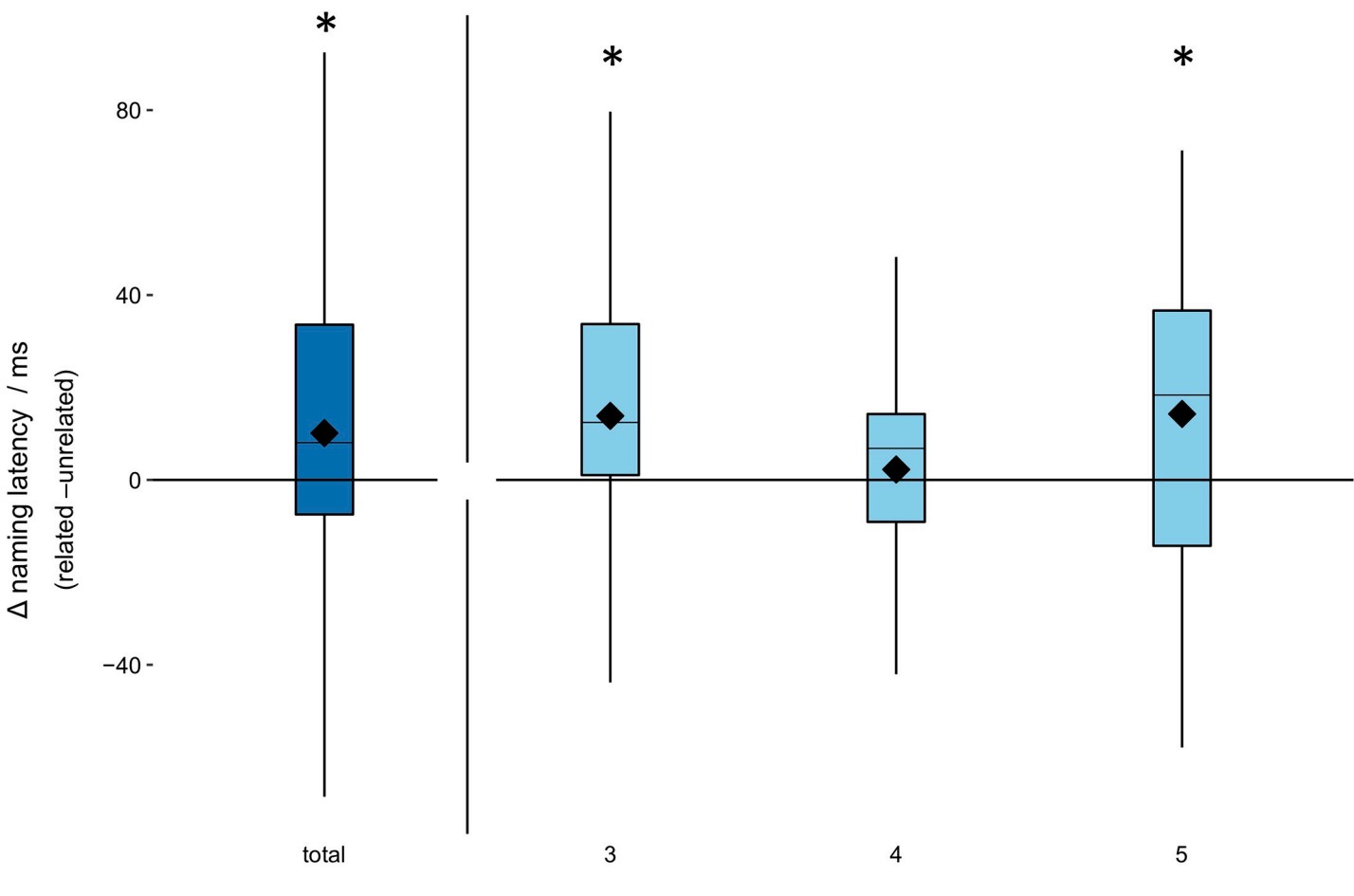

**Fig 2. Interference effect in total and across number of distractor words.** Total interference was significant at ~10ms. For 3 and 5 distractor words, interference was significant at ~15ms. There was no interference effect for 4 distractor words. Boxplots show mean, median, upper and lower quartiles and range.

effect and z-transformed. From the random structure correlation parameters as well as one contrast of the factor set size were removed to achieve convergence. As can be seen in Table 2, with this additional factor in the model, the main effect of picture type remained significant. However, it interacted (marginally) significantly with naming repetition, showing that the interference effect decreased across naming repetitions. When removing this interaction effect from the random structure for Item and Subject, the effect became highly significant (estimate = -9.50, $SE$ = 2.94, $t$ = -3.23, $p$ = 0.001). This means that participants as well as items varied with regard to this effect. Nevertheless, log likelihood tests showed that the more complex model fit the data better (logLik $\Delta X^2(2)$ = 22.08, p <0.001). We therefore report the more complex model. Overall RTs decreased by 39 ms on average for each additional target picture occurrence. The main effects of set size remained significant as well and did not interact with picture repetition (all t < 0.82, all p >0.414). Finally, the three-way interaction between picture TYPE, set SIZE and picture REPETITION was not significant (all t < 1.58, p > 0.114).

As can be seen in Fig 3, the interference effect is strongest at the first naming instance across all conditions. This was confirmed in a final (random intercept) model looking at the interaction of picture type and set size for the first naming instance. The effect of picture type was significant at ~44ms (estimate = 43.83, $SE$ = 9.01, $t$ = 4.86, $p$ < 0.001) and the interactions with

**Table 2. GLMM for the effect of picture type and set size across naming repetitions.**

| Term | Estimate | SE | t | p |
|---|---:|---:|---:|---:|
| Intercept | 870.81 | 5.83 | 149.48 | <0.001 |
| Picture type: rel-unrel [a] | 10.33 | 5.22 | 1.98 | 0.048 |
| Set size: 4–3 | -10.33 | 4.07 | -2.54 | 0.011 |
| Set size: 5–3 | -10.80 | 3.91 | -2.76 | 0.006 |
| Pic repetition | -39.29 | 6.24 | -6.29 | <0.001 |
| Pic type * set size: 4–3 | -11.87 | 3.98 | -2.98 | 0.003 |
| Pic type * set size: 5–3 | 0.48 | 5.57 | 0.09 | 0.932 |
| Pic type * pic repetition | -7.87 | 4.25 | -1.85 | 0.064 |
| Set size: 4–3 * pic repetition | -4.50 | 5.50 | -0.82 | 0.414 |
| Set size: 5–3 * pic repetition | 1.54 | 4.04 | 0.38 | 0.703 |
| Pic type * set size: 4–3* pic repetition | 7.86 | 4.97 | 1.58 | 0.114 |
| Pic type * set size: 5–3* pic repetition | -7.23 | 5.86 | -1.23 | 0.217 |

[a] henceforth "pic type".

set size were not significant (all t < 1.51, p > 0.132). This confirms a stable interference effect of around 44ms for all distractor conditions at the first naming instance.

## Eye tracking measures

To investigate viewing times of the mutually related and unrelated words in the word set, fixation durations were summed up on each AoI, yielding a total viewing time for each word in each trial. For each trial, total viewing time recorded by the eye-tracker was ~4100 ms on average (i.e. ~1900ms participants did not fixate on any of the AoIs or data were not recorded). The measure can be assumed to depend on data quality (e.g. blinks) and attentional resources. Fig 4 shows the mean viewing times for each related and unrelated word across all trials and participants, and for each lexical cohort condition [3, 4 or 5 mutually related words out of 8 words in total in each trial]. If there was no bias in fixating to members vs. non-members of the cohort, each word should be fixated for 1/8th of the total fixation time. The measures show that participants fixated longer on members than non-members, and therefore indicate the participant's categorization skills of semantically related and unrelated words in each word set.

The descriptive results were statistically confirmed by a GLMM with word type (related or unrelated) and distractor set size (i.e., 3, 4 or 5 related words) as fixed effects and a fully specified random structure.

Factor level contrasts showed that related words were fixated about 112ms longer than unrelated words (estimate = 114.44, SE = 16.05, t = 7.13, p < 0.001) and that the more related words there were, the shorter each word was fixated (4–3: estimate = -18.16, SE = 5.25, t = -3.46, p = 0.001; 5–3: estimate = -28.34, SE = 6.31, t = -4.49, p < 0.001). This did not depend on the type of word (related, i.e. part of the categorical distractor set, v.s. unrelated) that was fixated (no interaction effect, all t < 0.56, all p >0.574). For details see S3 Appendix, Table C1.

## Combined RT and eye tracking analysis

A final hypothesis concerned the relationship between fixation durations on the related words in the lexical cohort, and naming latencies for the consecutively named picture. We hypothesized that the longer participants fixated on the categorical distractor words within the cohort, the longer the RTs on naming a related picture would be. This relationship was analyzed by

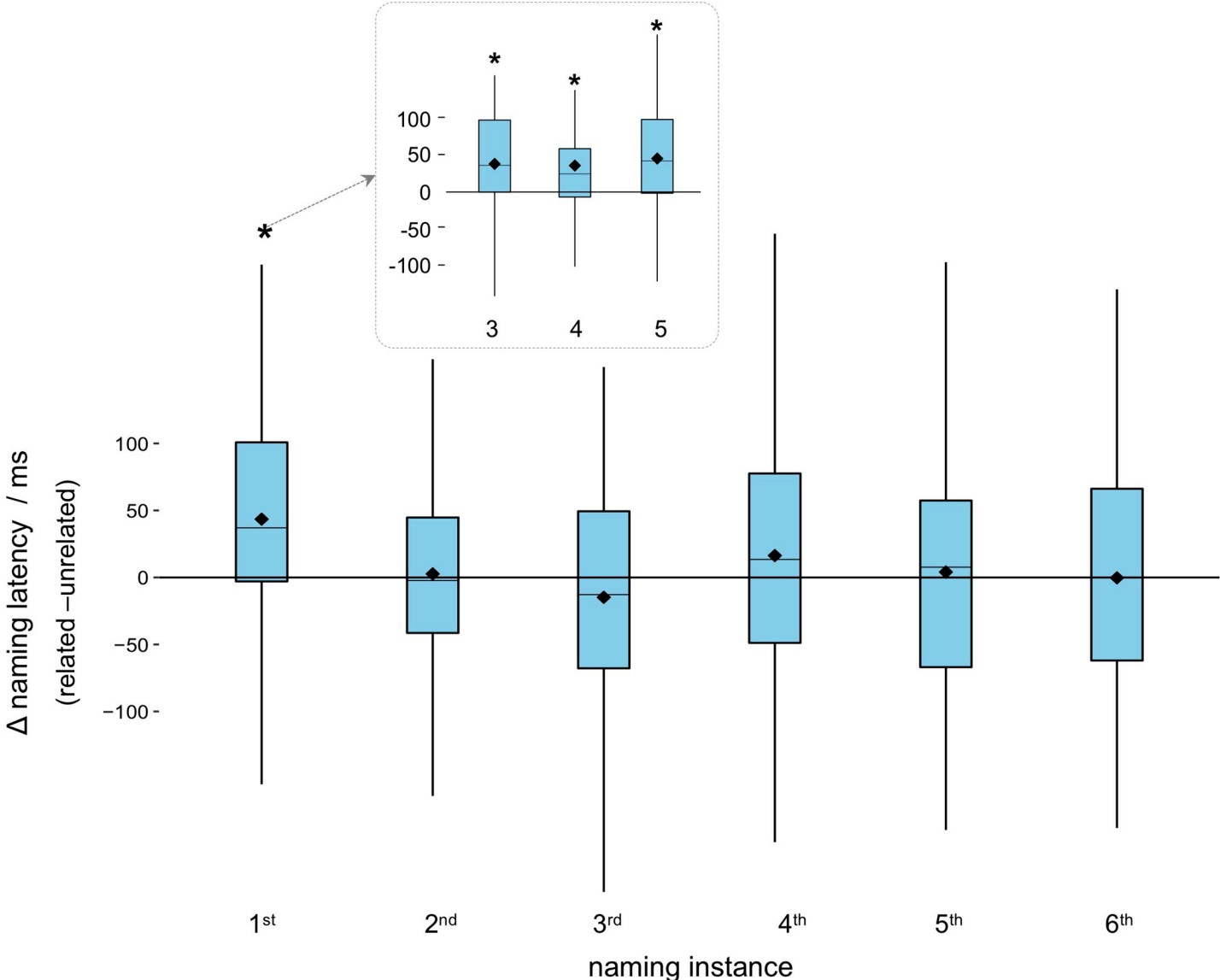

**Fig 3. Interference effect across naming instances (each picture was named 6 times in the related and unrelated conditions over the course of the experiment).** Interference was highly significant at the first naming instance and disappeared for the following repetitions. Note that the significant effect for the first naming instance was significant for all distractor conditions (inset). Boxplots show mean, median upper and lower quartiles and range.

another LMM adding fixation durations as a covariate (z-transformed) and a maximal random structure without correlation parameters. According to this model taking fixation durations into account, the interference effect in naming latencies remained marginally significant (estimate = 9.76, SE = 5.68, t = 1.72, p = 0.086). However, fixation durations did not influence naming latencies significantly (main effect of fixation durations: estimate = 2.54, SE = 2.58, t = 0.98, p = 0.327). Fixation durations also did not interact with picture type or set size (all t $<$. 1.56, all p $>$ 0.118). For details see S4 Appendix, Table D1.

This matches the results of Pearson's correlations between fixation durations and reaction times for each participant. The weak correlation became significant for 5 participants, but the average correlation coefficient was 0.

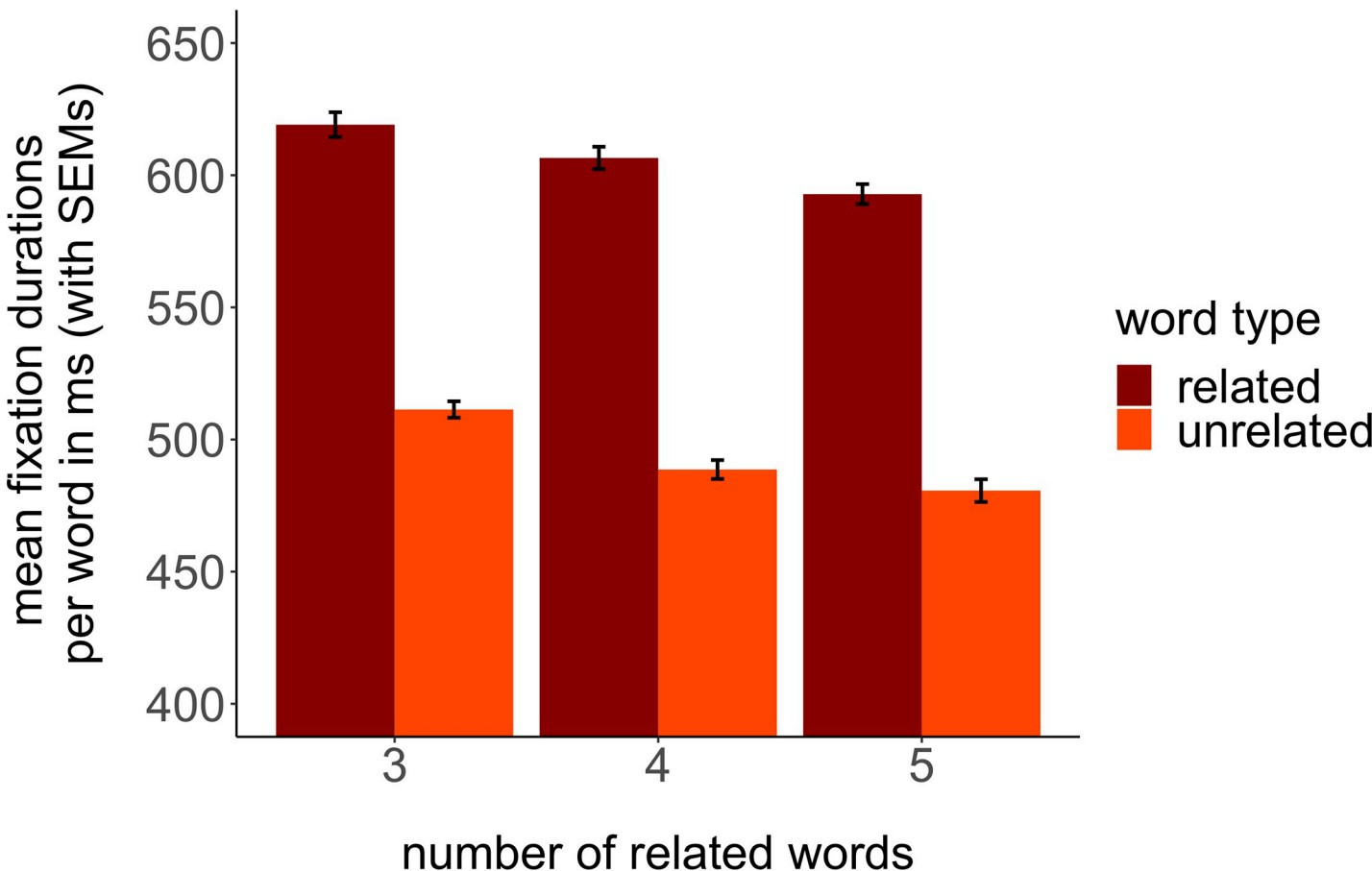

**Fig 4. Mean relative fixation durations (with SEMs) for each word as part of the distractor word set.**

## Discussion

In this study, we introduce a novel variation of the picture-word-interference (PWI) paradigm to investigate whether and how semantic interference effects through categorical distractors can be modulated. Consistently it has been shown that for the PWI paradigm categorically related single word distractors elicit slower naming responses for pictures from the same when compared to a different semantic category [see, e.g., 3,5,13,65]. This interference effect has been associated with the activation of a lexical cohort of related category members inducing competition during lexical selection and thereby delayed target word retrieval [13]. Besides lexical cohort effects, semantic relation between distractor words and the target word to be produced can also lead to facilitation. In that case the competition-induced slowing may be counteracted by effects likely arising at the conceptual level (i.e. 'animal with 4 legs'). To further elucidate the complex interplay between such opposing effects during picture naming we here address three questions using a variation of the PWI paradigm: First of all we address effects of the _size_ of the lexical cohort. Some evidence exists that activation is driven by category size, such that for larger categories (e.g., animals) more members/competitors can be activated when compared to smaller, narrower categories (e.g., insects); this relies on studies investigating semantic neighborhood density effects on picture naming [16,66]. Recent research, however, has also shown that interference increases for category members that are more closely related, leading to smaller numbers of exemplars [e.g., hoofed animals, 18,21]. In

the present study, we therefore manipulate lexical cohort activation in a more controlled way by changing the number of word distractors forming the semantic naming context for picture naming. While doing so, we kept the semantic categories in the stimulus set narrow, using categories that had elicited high interference effects in Rose et al. [18,21]. A second issue addressed in the present study is the question whether and how semantic _cohort recognition_ shapes semantic context effects. The prediction here is not straight forward: while semantic knowledge on the cohort is mandatory for the interference effect, the search for exemplars of the cohort in the visual word set may activate a conceptual, rather than a purely lexical search. While the latter should increase interference, the use of conceptual semantics is predicted to facilitate lexical access. To tap into this intriguing question, we used eye tracking to assess the individual processing of the semantic context. (iii) Finally, we address the question how effects of semantic context unfold across trials. This is of interest since continuous or blocked-cyclic naming paradigms suggest a build-up of interference with repeated exposure, while PWI-paradigms typically do not report sequence effects.

Beyond the more general inquiry into how semantic context shapes picture naming, we specifically ask whether an increased number of categorically related words in a PWI paradigm inhibits or facilitates retrieval of a target picture name and whether this effect changes over repeated instances of naming. Analyzing fixation times, we additionally assess rather than assume semantic 'competence' for the different conditions.

In brief, our findings confirm an interference effect when words categorically related to the picture to be named are presented prior to the picture. This effect, however, disappears with repeated naming and the duration of fixation on the semantically related distractor words does not predict naming latency. Most notably the effect of the number of semantically related words in the distractor set is contrary to predictions based on a simple interference account. With an increasing number of categorically related words in the distractor set, semantic interference did not increase further.

The increase in naming latency when words presented prior to the picture are categorically related replicates previous results using semantic PWI. The interference effect is generally interpreted to show that reading the words activates lexical representations connected through one category node, making them strong enough competitors to inhibit target selection when the target was part of the same semantic category [see e.g., 67,68, on the time course of this process]. Replicating this finding in our novel paradigm indicates that interference of categorically related words with the naming of a picture is robust even if timing of the individual trial and the number of distractors is substantially altered. The overall effect of around 10 ms is smaller than found in typical PWI paradigms, but is statistically significant across all participants and trials, even when taking participant and stimulus variation into account using mixed effects modeling.

Notably, however, the development of this effect over repetitions reveals that a net-interference effect occurred only at the first out of six naming instances for each target picture for which it was much larger (~44 ms), irrespective of the number of categorically related items in the distractor set. The effect dissipates across the remaining target presentations, and overall reaction times decrease by about 120 ms from first to last naming instance of each target picture, suggesting an increase in facilitatory mechanisms, neutralizing interference effects. Such a reduction of the interference effect evidenced by naming latencies when comparing repeated naming instances has not been demonstrated for PWI paradigms [and see 47 for evidence that interference remains stable across naming repetitions of the same picture]. It also stands in contrast to findings from the blocked cyclic or continuous naming paradigms, where reaction times increase across trials [cumulative interference, e.g., 12], or interference only appears from the second presentation cycle onwards and afterwards remains stable or even increases

slightly [8,9]. Note, however, that this sort of cumulative interference results from the repetition of *categories*, not single *items*. Nevertheless, our findings also contrast with alternative explanations of the origins of semantic interference, specifically the response-exclusion hypothesis [24]. This theory posits that through frequent exposure, task-relevant responses (e.g. names of pictures from the same semantic category) need to be actively excluded from an articulatory output-buffer resulting in delayed naming. In our paradigm, these task-relevant items would include previously named pictures, and previously fixated words that were part of a category word set. But as we discuss in more detail below, our results show that indeed frequent exposure to the material leads to *faster* naming, thus making an explanation of an effortful and therefore inhibitory monitoring mechanism unlikely.

The most noteworthy finding of the current study pertains to the effect of distractor set size: Contrary to our hypothesis, interference did not increase from 3 to 4 or 5 distractor words that were semantically related to the picture, but was equally strong (~15 ms) for 3 and 5 distractor words. Interestingly, when 4 distractors were part of the word set, naming was not interfered at all. In sum, regarding the extent of activation modulated by cohort size of semantically distractors we have to reject the hypothesis that a larger number of distractor words induces more competition on target word retrieval. This will be further discussed below.

Using eye-tracking, a third relevant finding relates to how participants process the semantic context provided by the distractor words: average fixation time on categorically related words was significantly longer compared to that on the remaining, unrelated words. The finding is notable in two ways: firstly, it confirms that neurotypical participants implicitly categorize words without specific instruction to do so. Moreover, the analysis of the eye-tracking data allowed for correlating fixation time on the semantically related exemplars in the distractor word set with naming latencies for pictures from the respective semantic category. Contrary to the assumption that longer and thereby more intense processing might lead to larger interference, the correlation was around zero for all participants. Hence, we find no indication that processing distractor words longer increases interference. If longer fixation elicits stronger lexical activation an increase in naming latency would be expected. Our null results indicate that some facilitatory effect counteracts such a purely lexical competition effect.

The fact that we find no evidence for the expected increase in the interference effect for naming latencies with an increasing number of distractor words requires discussion. A closer look at distractor conditions across naming repetitions revealed that this global result was influenced by an interaction with repetition. At the first naming instance, there was equally strong interference for all distractor conditions of around 44 ms. For all future naming instances however, interference disappeared or even turned into facilitation (= faster naming latencies for related compared to unrelated pictures). It should be noted that across the 252 trials the overall 42 'items' appeared 54 times (3 times as related, 3 times as unrelated pictures to be named and additionally 24 times as related, 24 times as unrelated distractor words). We argue that the very substantial effect of overall familiarization with the set of items (latency decrease of 120 ms over the course of the experiment) is not dependent on the number of related distractor words and holds for related and unrelated conditions.

A long stimulus onset asynchrony (SOA) and a strong familiarization with the stimulus set, both novel features of our paradigm, will have improved prediction of the target item and promoted a rather conceptual than purely lexical activation of category members. As opposed to typical single-word PWI paradigms, in our novel paradigm a negative stimulus onset asynchrony (SOA) of 6 s was used, which is much longer than in the typical single-distractor-word paradigms. Indeed, previous studies have shown that SOAs of -1000 ms or -400 ms led to facilitation rather than semantic interference for categorical distractor words presented prior to the picture [36,38]. We used the long SOA to ensure that each word, especially from the

categorically related distractor words, was fixated and processed. On average participants fixated ~500 ms on each word belonging to the respective lexical cohort in the distractor set. Our results show that even with this long SOA, substantial semantic interference was elicited at the first naming instance. However, we suggest that, together with the cumulative exposure to the stimuli, this long SOA enhanced the implicit analysis of conceptual features of the lexical cohort, counteracting lexical competition. This conceptual analysis is also reflected by reduced fixation durations per word when more categorically related words were presented, and is consistent with the SLN account [13,14], in which priming on the conceptual level leads to facilitation of lexical retrieval. We therefore propose that a complex interplay between lexical interference and semantic priming effects is causal for our findings, whereby frequent exposure to the stimulus material elicits a facilitative effect on naming latencies, counteracting interference.

## Outlook and implications for future research

The paradigm we have introduced in this study provides important information on the nature of picture-word interference and the processing of semantic context. Results suggest that a larger number of distractors not necessarily increases interference, even though previous research had suggested this outcome [27]. Long SOAs and frequent repetition of the stimulus material are candidate factors to lead to increased facilitation abolishing the initially robust interference effect. Furthermore, more evidence is needed to understand the relationship between semantic competence and naming latencies. In the present study, participants' semantic competence was unimpaired, and this was reflected by their ability to categorize the mutually related words in the word set. So far it is unclear how impaired semantic competence interacts with the semantic interference effect. Research on the language disorders in participants with semantic memory deficits such semantic variant Primary Progressive Aphasia (svPPA) has indicated continuing loss of semantic features as the underlying mechanism to progressive naming impairments [69–72]. This might lead to the inability to distinguish categorically related and unrelated members of the word set, and therefore to reduced or absent interference effects. The combination of our variation of the PWI paradigm with eye tracking therefore seems an apt tool to examine this phenomenon in clinical populations with (e.g. svPPA) and without (e.g. Broca's Aphasia) impairments of the semantic system.

## Conclusion

In the current study we put forward a new paradigm to investigate influences of semantic context on word retrieval. We stipulated that semantic interference effects consistently found for classical PWI paradigms could be modulated in a variation of the paradigm. Here, instead of one distractor word, several distractors were presented at once, in form of a circle. This allowed us to examine the processing intensity of semantic context and parametric manipulations of the number of distractor words from on semantic category. We have demonstrated that multiple distractor words from one semantic category elicit interference–similar to that in classical one-word interference paradigms but that this effect is present only the first time a picture is presented, where it is independent of distractor set size. It then dissipates across repetitions, mediated by facilitative processes leading to faster lexical access. Moreover, interference did not increase for a larger cohort of distractor words. These findings suggest a complex interaction between activation on the lexical and conceptual processing level, which depends on lexical cohort size as well as frequency of exposure to the semantic context across repetitions within the experiment.

## Supporting information

**S1 Fig. Quantile-quantile plots showing distribution of RTs.** Panel A: raw RTs, panel B: RTs with gamma distribution.
(TIFF)

**S2 Fig. Naming latencies (raw means and SEMs) across naming instances (each picture was named 6 times in the related and unrelated conditions over the course of the experiment).**
(EPS)

**S1 Appendix. Stimuli.**
(DOCX)

**S2 Appendix. Naming latencies.**
(DOCX)

**S3 Appendix. Fixation durations.**
(DOCX)

**S4 Appendix. Combined analysis of reaction times and fixation durations.**
(DOCX)

## Acknowledgments

We would like to thank Dr Luke Tudge at Berlin School of Mind and Brain (Humboldt-Universität zu Berlin, Germany) for his advice on statistics in the paper.

## Author Contributions

**Conceptualization:** Cornelia van Scherpenberg, Rasha Abdel Rahman, Hellmuth Obrig.

**Data curation:** Cornelia van Scherpenberg.

**Formal analysis:** Cornelia van Scherpenberg.

**Funding acquisition:** Cornelia van Scherpenberg.

**Investigation:** Cornelia van Scherpenberg.

**Methodology:** Cornelia van Scherpenberg, Rasha Abdel Rahman, Hellmuth Obrig.

**Project administration:** Cornelia van Scherpenberg, Hellmuth Obrig.

**Resources:** Hellmuth Obrig.

**Supervision:** Rasha Abdel Rahman, Hellmuth Obrig.

**Validation:** Cornelia van Scherpenberg, Hellmuth Obrig.

**Visualization:** Cornelia van Scherpenberg, Hellmuth Obrig.

**Writing – original draft:** Cornelia van Scherpenberg.

**Writing – review & editing:** Cornelia van Scherpenberg, Rasha Abdel Rahman, Hellmuth Obrig.

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
