## [Decision Letter · Decision Letter 0]

26 Sep 2019

PONE-D-19-20412

A novel multi-word paradigm for investigating semantic context effects

in language production

PLOS ONE

Dear Ms van Scherpenberg,

Thank you for submitting your manuscript to PLOS ONE.  Let me begin by apologizing for the unusually long turnaround time on your manuscript.  There were a series of limited delays at various stages of the manuscript handling process, which in their culmination resulted in a more significant delay overall.

After careful consideration, we feel that your manuscript has merit but does not fully meet PLOS ONE’s publication criteria as it currently stands. Therefore, we invite you to submit a revised version of the manuscript that addresses the points raised during the review process, in line with the comments below.  

Two expert reviewers have weighed in on the manuscript, and I have read it carefully myself.  Both of these reviewers are fairly critical of the manuscript, expressing some difficulty in seeing how the conclusions can be seen to receive clear support from the data (a pivotal criterion for publication in PLOS ONE).  For one, both reviewers point to issues regarding the coverage and grounding of the theoretical background, which compromised their ability to fully evaluate the conclusions of the work.  Reviewer 1, for instance, asks for more clarity on why contextual richness was manipulated by increasing the number of, rather than degree of relatedness of, the semantically related distractors, and what motivates the particular choices for number of distractors used in light of previous research.  Reviewer 2 points to some respects in which the background is incomplete, and suggests that the main contributions of interest depend on the research being situated in an expanded characterization of current theoretical discussions.  

Both reviewers also question various aspects of the predictions and the way that the results are interpreted against them.  For instance, Reviewer 1 requires further information to be convinced that the null effects found can be interpreted as the result of offsetting facilitation and interference effects.  Further, both reviewers have significant questions regarding the Eye-Mind hypothesis; for instance, both wonder why participants should fixate longer on words which they have recognized to belong to the same category.  Reviewer 2 further wonders why one would predict that longer fixation times on related words belonging to the lexical cohort would be predicted to induce stronger competition that results in inhibition on the target.  This reviewer also asks whether lexical frequency was controlled for in light of previously established results, and seeks clarification on how fixation time on a word can be treated as independent of the number of related words available.

Whereas Reviewer 1's comments do not provide a rationale for further consideration, Reviewer 2 finds merit in the work's attempt to combine different naming paradigms to disentangle the nature of the semantic effects in language production.  In view of this, I would consider a revised version of your manuscript, if you're inclined to resubmit after taking on a thorough consideration of the feedback from both reviewers.  In line with the concerns expressed above and in the reviews, and in light of PLOS ONE standards, my focus in evaluating a resubmission would be on enduring that the experiments and statistics are well-explained and carried out to a high technical standard, that the predictions are clear and theoretically motivated, and that the conclusions are well-supported by the data.  Whereas you are free to further address the novelty and potential impact of your results, those considerations will not be determinative of the final outcome.  As it appears that a fair bit of reworking may be required, I'm naturally disinclined to make a prediction about what that outcome is likely to be at this time.  I would ask the same two reviewers to evaluate a resubmission, only recruiting new ones if one or both were to decline.

We would appreciate receiving your revised manuscript by Nov 10 2019 11:59PM. To enhance the reproducibility of your results, we recommend that if applicable you deposit your laboratory protocols in protocols.io, where a protocol can be assigned its own identifier (DOI) such that it can be cited independently in the future. For instructions see: http://journals.plos.org/plosone/s/submission-guidelines#loc-laboratory-protocols

We look forward to receiving your revised manuscript.

Kind regards,

Andrew Kehler, Ph.D

Academic Editor

PLOS ONE

Journal Requirements:

1. We note that you have stated that you will provide repository information for your data at acceptance. Should your manuscript be accepted for publication, we will hold it until you provide the relevant accession numbers or DOIs necessary to access your data. If you wish to make changes to your Data Availability statement, please describe these changes in your cover letter and we will update your Data Availability statement to reflect the information you provide.

2. Please remove your figures from within your manuscript file, leaving only the individual TIFF/EPS image files, uploaded separately.  These will be automatically included in the reviewers’ PDF.

Reviewers' comments:

Reviewer's Responses to Questions

**Comments to the Author**

1. Is the manuscript technically sound, and do the data support the conclusions?

Reviewer #1: No

Reviewer #2: Yes

2. Has the statistical analysis been performed appropriately and rigorously? 

Reviewer #1: Yes

Reviewer #2: Yes

3. Have the authors made all data underlying the findings in their manuscript fully available?

Reviewer #1: No

Reviewer #2: No

4. Is the manuscript presented in an intelligible fashion and written in standard English?

Reviewer #1: Yes

Reviewer #2: Yes

5. Review Comments to the Author

Reviewer #1: The authors present a modification of the standard picture-word interference (PWI) paradigm. In the standard PWI paradigm, a single-word distractor is displayed in the center of and at the same time with a target picture to be named. Here, the authors displayed eight-word distractors aligned in a circle around a target picture six seconds before the picture presentation (to facilitate distractor inspection). Two further changes included varying the number of semantically related (SR) distractors (between three and five of the eight distractor words were SR to the target on any given trial) and repeating distractor-target sets six times in the SR condition, and six times in the unrelated (UR) condition. The authors tested whether increasing “semantic context richness” (via 3,4, or 5 SR word distractors), increasing “processing intensity” (how much Ss paid attention to the SR words as measured by eye gaze fixations), and repeating distractor-target pairings increased the magnitude of semantic interference. Subjects (n=24) were significantly slower on average to name pictures in the context of SR (3-5 words) vs. UR words (8 words; the main effect was not significant across all analyses). Contrary to predictions, there were no effects of increasing numbers of SR distractors nor increasing SR word distractor gaze durations on the magnitude of semantic interference. In contrast to effects of repetition described in continuous and blocked-cyclic naming semantic interference paradigms (cf. Howard et al., 2006; Kroll & Stewart, 1994), repetitions did not increase the magnitude of semantic interference but instead eliminated it after the first repetition. The authors interpret the lack of interactions with the semantic interference effect as due to a “complex interaction between activation on the lexical and conceptual processing level” (p. 33, line 567). Authors interpret the lack of semantic interference after the first naming repetition being due to the influence of semantic facilitation.

Overall impression

The question of how semantic context affects word production by speeding it up or slowing it down continues to be one that is still unresolved in the language production literature. The introduction of a potentially novel paradigm which would help answer this question is a theoretically important endeavor. Unfortunately, the paper does not sufficiently describe the theoretical motivations for the specific PWI paradigm modifications introduced. The lack of theoretical grounding in the introduction and the general discussion combined with mostly null results makes it difficult to understand the utility of the new paradigm, the implications for previous work, or the implications for current theory concerning the mechanics of word production. Thus, as currently presented, conclusions are not supported by the data and the work does not meaningfully contribute to the base of scientific knowledge.

Major comments

1. One of the goals of the paper was to modify the richness of the semantic context. However, instead of using the degree to which targets and distractors are semantically related, richness was altered by increasing the number of SR items. It was unclear why this modification was adopted and how it compares theoretically and empirically to previous work manipulating the degree of semantic overlap. Further, it was unclear why eight word distractors were included (and not four, for example), or why the number of SR distractors varied from 3 to 5, as opposed to something else.

2. The impact of repetition on the magnitude of semantic interference is difficult to interpret. The authors interpret the null effect between repetition (time points 2-6) and the semantic interference effect as being a result of the summation of facilitation and interference. To support this interpretation, please provide a prediction how RTs in the SR vs. UR condition across repetitions should occur to support the explanation, and then present the results in both conditions across repetitions. Second, please motivate at what level in the production system repetition effects occur, and how they interact with interference effects. Lastly, please frame these results with reference to previous investigations of semantic interference/facilitation effects during word production.

3. Introduction, p. 11 line 116: Please provide more explanation concerning the “eye-mind” hypothesis and the rationale behind why participants should fixate longer on words that belong to the same semantic category within particular paradigms. Does word fixation depend on the task the participant was instructed to do? What does semantic competence refer to? Relatedly, please provide clearer explanation as to why the different degrees of word fixation for related vs. unrelated words was due to “categorization” skills and not some other factor, given that Ss were instructed only to “inspect” the word distractors (p. 17, line 251; p. 24, line 378; cf. p. 30, line 491).

Minor comments

1) Reference #13 is incomplete.

2) Please explain why it is interesting that interference effects are stable in auditory PWI with phonological distractors (p. 12 line 159).

Reviewer #2: Two very well-documented empirical findings on the field of lexical access in speech production are the following. The first phenomenon refers to the fact that semantic context modulates the retrieval of words from the mental lexicon. The second one is the observation that the direction of this semantic context effect (facilitation or interference) depends on at least two factors; 1) the experimental task (e.g., PWI, blocking naming/cyclic naming, continuous naming); and 2) the semantic relationship between the target response and the prime element (i.e., coordinate words, superordinate words, associate words…).

Van Scherpenberg, Abdel Rahman and Obrig states that the main aim of their study was “to test whether semantic interference is confirmed for this novel paradigm, and whether the number of related words modulates its magnitude” (abstract); / “In the present study we focus on categorical semantic relations and investigate whether manipulating the extent of lexical activation within a lexical cohort modulates inhibition on subsequent picture naming.” (page 5). I wonder whether we really need new evidence showing that lexical access in language production is affected by semantic context and that this modulation can be incremental (e.g., Howard et al. 2006). Sincerely, I don’t think this aim would deserve publication in a journal as Plos ONE

In my opinion, the main interest of the current manuscript relies on the attempt to combine different naming paradigms to disentangle the nature of the semantic effect(s) in language production. To this respect, the topic discussed is timely and fits well with recent discussions in the field about the trade-off between accuracy and speed in picture naming (e.g., Nozari & Hepner, 2018). At the same time, the combined methodology of the study allows the comparison between word processing (eye tracking measure) and lexical access (naming latency measure) within the same task. In sum, the multi-novel experimental paradigm the authors adopt can be very interesting only if it is discussed in the context of current and relevant theoretical discussions.

In my opinion however, there are some issues that need to be clarified.

1-Theoreical background (literature review – scant)

The introduction is a little bit confusing and theoretical biased. As mentioned above, semantic effects (facilitation and interference) have been reported mainly in three experimental tasks (PWI, blocking/cyclic naming and continuous naming). In the first page of the Introduction (page 3), the authors focus on one model, the Swing Lexical Network (SLN), and introduce the semantic context effects in relation to this model. In particular, the SLN approach can explain facilitation and semantic effects in (all) experimental tasks depending on the trade-off between conceptual priming and lexical competition. Of course, I have no problems with this choice; but see Navarrete et al (2016) for a review of the same topic based on the distinction between inter-level trial semantic manipulation tasks (PWI) and intra-level semantic manipulation tasks (cyclic and continuous naming) without reference to any specific theoretical model.

In addition, the authors need to make explicit at certain point in the introduction the following aspects:

1a-while the SLN assumes that the balance between two mechanisms (conceptual priming and lexical competition) is able to explain the diverse pattern of semantic effects, other proposals have made explicit the claim that different tasks entails different cognitive mechanisms to resolve lexical retrieval in semantic context. For instance, Mahon and colleagues (2007) have argued that semantic interference in the PWI can be explained by a post-lexical mechanism of self-monitoring (see also Dhooge & Hartsuiker, 2011); while semantic interference in picture naming tasks would arise at the semantic-to-lexical connections (through an inhibition mechanism that weakens these connections) (Navarrete et al. 2010; 2014). The authors need to mention this alternative(s) explanation(s).

1b-The experimental set entails the presentation of two events in this order: 1-the presentation of an array with several written words, and 2-the presentation of a picture target to be named. Previous literature has already explored to which extend written word processing influences the successive picture naming event. Again, further literature has to be included and mentioned:

Belke (2013); Navarrete et al. (2013); Vitkovitch et al., (2010); Vitkovitch & Cooper (2012)

1c-(Page 6; lines 18-148). The interaction between semantic context and cycle observed in the cyclic naming tasks (facilitation first cycle, interference from the second cycle onward) has been extensively explored by Navarrete et al 2014. This has to be acknowledged.

2-Experimentnal predictions (unclear)

2a-Page 5. Eye-mind hypothesis. The authors need to explain the “eye-mind hypothesis”. What does it means that participants should fixate on words longer which they have recognized to belong to the same category? Why? I would predict the reverse effect, less fixation time because of the semantic priming between related words

2b-Page 5. Eye-mind hypothesis. According to one of the references cited by the authors (Rayner, 1995), word frequency is one of the most critical variable determining the amount of time a word is fixed. Did the authors control the lexical frequency between the 8 words of the arrays? I guess that the amount of time I can fixate the words cat-dog-horse would depend on their own lexical frequency, as well as on the lexical frequency of the other words in the array. For instance, if in an array the non-related words are high frequency words (e.g. house-car-table-hand) I would have more/less time to fixate the related words (e.g., cat-dog-horse) than if the non-related words are low frequency words (e.g., skyscraper-tractor-stool-chin)

2c-Page 5. Eye-mind hypothesis. We are informed that: “We predict that the longer participants fixate on related words belonging to the lexical cohort, the more activity will spread to this cohort and induce stronger competition resulting in inhibition on target”. Justify this assumption

2d-The lack of interaction between Type array (related vs. unrelated) and Size (3, 4, 5 related words) seems problematic with the prediction that fixation times would increase according to the number of related words in the array. If I understood correctly: the authors predict that the average of fixation time for a related word (cat) should be longer when the word appears together with 4 related words (horse-pig-rabbit-dog) than when two related words (horse-pig). The problem is that the average-time a participant is looking at one word is not independent of the time the participant dedicates to look at the other words. Sorry if I miss something here!!!

Just to illustrate my point (arrays are presented for 6000 ms as in the experiment). If I fixate in a Size 3 – lexical cohort the word cat for 1000 ms, then the maximum fixation time I can dedicate to look at the other words of the array is 5000 ms; and for instance I look at the other two related words 2000 ms (horse 1000 ms and pig 1000ms). But if I fixate cat for 2000 ms in a Size 5 – lexica cohort, then the total time I can fixate the other words of the array is 4000 ms; and for instance I look at the other four related words 1000 ms (horse 250 ms, pig 250 ms, rabbit 250 ms, dog 250 ms). This hypothetical scenario serves to illustrate the point that, under the authors' assumption that fixation time depends on semantic context, the average fixation time for a single word cannot be independent of the semantic Size – lexical cohort! Therefore, it’s unclear what we can learn from the fixation measure. The authors need to clarify this confound.

Minor points:

Statistics. Are the reported beta valuescorrect? Maybe they should be b values?!?

References:

Howard, D., Nickels, L., Coltheart, M., & Cole-Virtue, J. (2006). Cumulative semantic inhibition in picture naming: Experimental and computational studies. Cognition, 100(3), 464-482.

Nozari, N., & Hepner, C. R. (2018). To select or to wait? The importance of criterion setting in debates of competitive lexical selection. Cognitive Neuropsychology, 1-15.

Navarrete, E., Mahon, B. Z., Lorenzoni, A., & Peressotti, F. (2016). What can written-words tell us about lexical retrieval in speech production?. Frontiers in Psychology, 6, 1982.

Mahon, B. Z., Costa, A., Peterson, R., Vargas, K. A., & Caramazza, A. (2007). Lexical selection is not by competition: a reinterpretation of semantic interference and facilitation effects in the picture-word interference paradigm. Journal of Experimental Psychology: Learning, Memory, and Cognition, 33(3), 503.

Dhooge, E., & Hartsuiker, R. J. (2011). The distractor frequency

effect in a delayed picture-word interference task: Further

evidence for a late locus of distractor exclusion.

Psychonomic Bulletin and Review, 18, 116–122.

Navarrete, E., Mahon, B. Z., & Caramazza, A. (2010). The cumulative semantic cost does not reflect lexical selection by competition. Acta psychologica, 134(3), 279-289.

Navarrete, E., Del Prato, P., Peressotti, F., & Mahon, B. Z. (2014). Lexical selection is not by competition: Evidence from the blocked naming paradigm. Journal of Memory and Language, 76, 253-272.

Belke, E.(2013).Long-lastinginhibitorysemanticcontexteffectsonobjectnaming are necessarilyconceptuallymediated:implicationsformodelsoflexical- semanticencoding. J. Mem.Lang. 69, 228–256.doi:10.1016/j.jml.2013.05.008

Vitkovitch, M., Cooper-Pye, E., & Ali, L. (2010). The long and the short of it! Naming a set of prime words before a set of related picture targets at two different intertrial intervals. European Journal of Cognitive Psychology, 22(2), 161-171.

Vitkovitch, M., & Cooper, E. (2012). My word! Interference from reading object names implies a role for competition during picture name retrieval. The Quarterly Journal of Experimental Psychology, 65(6), 1229-1240.

Rayner, K. (1995). Eye movements and cognitive processes in reading, visual search, and scene perception. In Studies in visual information processing (Vol. 6, pp. 3-22). North-Holland.

6. PLOS authors have the option to publish the peer review history of their article (what does this mean?). If published, this will include your full peer review and any attached files.

Reviewer #1: No

Reviewer #2: No

---

## [Author Response · Author response to Decision Letter 0]

10 Dec 2019

Reviewer 1

Comment 1: 

One of the goals of the paper was to modify the richness of the semantic context. However, instead of using the degree to which targets and distractors are semantically related, richness was altered by increasing the number of SR items. It was unclear why this modification was adopted and how it compares theoretically and empirically to previous work manipulating the degree of semantic overlap. Further, it was unclear why eight word distractors were included (and not four, for example), or why the number of SR distractors varied from 3 to 5, as opposed to something else.

Our response: 

We thank the reviewer for the comment and hope to now avoid the apparent imprecision of terminology. Our aim in the current paper was to investigate a parametrical manipulation of the classic semantic interference effect found in picture-word interference (PWI) paradigms. As discussed below in response to the constructive criticism of reviewer 2, the novel combination of different paradigms used in the research on semantic context effects was another rationale underlying the design of the paradigm introduced in the current paper. Regarding the aim to parametrically vary semantic interference we wanted to manipulate the activation level of the lexical cohort leading to competition between lexical entries and therefore interference during target selection, as suggested by the Swinging Lexical Network account (Abdel Rahman & Melinger, 2009, 2019). This indeed is different from the approach to modulate the “richness” of the semantic context, as has been done in previous studies (e.g., Rabovsky, Schad, & Abdel Rahman, 2016; Rose, Aristei, Melinger, & Abdel Rahman, 2018). We therefore chose to develop a novel paradigm that allows implementing this manipulation, while at the same time monitoring implicit semantic processes by using eye tracking. Our rationale was that in order to investigate the processing differences of categorically related vs unrelated words with eye tracking, we need to present more than two words of each cohort. With allowing for incremental changes of numbers of distractors, and balancing the number of categorical vs unrelated stimuli across trials, this lead to the design that we present here, where eight distractor words are arranged in circle. We hope this explanation clarifies the reviewer’s concerns about our study design. To avoid the imprecision apparently suggested by our previous version we altered the respective sentences in the introduction (line 57 ff.) “… by changing the intensity of semantic context activation…”

Comment 2:

The impact of repetition on the magnitude of semantic interference is difficult to interpret. The authors interpret the null effect between repetition (time points 2-6) and the semantic interference effect as being a result of the summation of facilitation and interference. To support this interpretation, please provide a prediction how RTs in the SR vs. UR condition across repetitions should occur to support the explanation, and then present the results in both conditions across repetitions. Second, please motivate at what level in the production system repetition effects occur, and how they interact with interference effects. Lastly, please frame these results with reference to previous investigations of semantic interference/facilitation effects during word production.

Our response:

We thank the reviewer for these questions and suggestions. Based on information from other paradigms like the blocked cyclic or continuous naming paradigms, our hypothesis was that for related pictures, naming latencies should increase across repetitions. This hypothesis does not hold for unrelated pictures. We analysed exactly this effect of repeated presentation on related and unrelated items. The respective results are visualized in the figure attached (Supplementary Figure). The results are contrary to our hypothesis in that there clearly is no increase in interference for the related pictures. Interestingly however the difference between unrelated and related items wanes after the first presentation. This effect is reported in the current manuscript and potential explanations are provided. Another effect is the general decrease in naming latencies for both conditions, which is also confirmed by statistics. Since we did not have a hypothesis regarding this overall ‘learning’ effect we decided to not include the result in the paper. 

A major reason beyond the fact that we did not make any predictions regarding an overall repetition effect is that we feel the paper and the discussion of the results is rather complex already without including this finding. However, if the reviewer believes it would be of note to potential readers we would include the graph illustrating both conditions over repeated naming instances. Explanations for this general learning effect are certainly very tentative, since a similar design has not been tested as yet. Alternatively, one might add a sentence regarding the effect stating that this effect was not targeted and is therefore not discussed further. While we still believe the paper would be more readable and clearer regarding our main hypotheses without the presentation of the results, we are grateful for an opinion, which we are happy to follow. 

Supplementary Figure: Effect of naming latencies across repetitions for both related and unrelated naming conditions.

Comment 3:

Introduction, p. 11 line 116: Please provide more explanation concerning the “eye-mind” hypothesis and the rationale behind why participants should fixate longer on words that belong to the same semantic category within particular paradigms. Does word fixation depend on the task the participant was instructed to do? What does semantic competence refer to? Relatedly, please provide clearer explanation as to why the different degrees of word fixation for related vs. unrelated words was due to “categorization” skills and not some other factor, given that Ss were instructed only to “inspect” the word distractors (p. 17, line 251; p. 24, line 378; cf. p. 30, line 491).

Our response:

We agree with the reviewer that more information is needed to justify our claims. We have rephrased this paragraph (Introduction, lines 133 ff.) to (1) better illustrate the theoretical value of the eye-mind hypothesis for our study, and to (2) explain what we mean by semantic competence.

(1) The eye-mind hypothesis refers to the immediate relationship of fixation durations and processing intensity in reading and is the theoretical basis for our hypothesis that longer fixation durations to semantically related items should lead to stronger interference effects.

(2) Eye tracking has been used to assess semantic competence / semantic deficits in clinical populations. We hypothesize that our paradigm is suited to do so as well and that longer fixation durations to semantic category members compared to non-members indicate intact semantic processing abilities, i.e., semantic competence.

We hope that the paragraph addresses the reviewer’s concerns and makes our assumptions clearer. 

Introduction, lines 133 ff.:

We proceed from the rationale that eye tracking can be used to investigate the semantic ‘competence’ of viewers. This assumption rests on paradigms performed in people with Primary Progressive Aphasia (PPA) and neurotypical controls. Suggesting impaired semantic memory abilities, participants suffering from PPA (Faria, Race, Kim, & Hillis, 2018; Seckin et al., 2016) fixated on semantically unrelated objects (foils) more often and longer when compared to neurotypical controls, likely indicating difficulties to establish the semantic relationships between concepts. Here we hypothesize that the neurotypical young adults, are semantically competent, and should hence fixate on words longer which they have recognized to belong to the same category, when compared to the unrelated words. Thus analysis of fixation times was used to investigate the semantic ‘competence’ of the participants in each trial. Additionally we can use this measure to estimate the extent to which they activate the lexical cohort. According to the eye-mind hypothesis (Just & Carpenter, 1980; Rayner, 1995), readers’ gaze durations are immediately linked to what they are processing. That is, words that are fixated longer are also processed longer. We therefore predict that the longer participants fixate on related words belonging to the lexical cohort, the more activity will spread to this cohort and induce stronger competition resulting in inhibition on target selection.

-----

Although the data are not yet fully analyzed, it may be of interest, that we have recently tested a cohort of people with a chronic acquired brain lesion in the left hemispheric extended language network with the paradigm introduced in the current paper. Preliminary analyses support our rationale to “operationalize” gaze preference as an indicator of lexico-semantic competence. What we find (with the caveat of preliminary analysis) is that this measure varies strongly across patients from no preference for related vs. unrelated items to patterns which are largely similar to our findings in the neurotypical group reported on here. 

Reviewer 1 minor comments:

Reference #13 is incomplete.

Our response:

We thank the reviewer for this note and have completed Reference #13.

Please explain why it is interesting that interference effects are stable in auditory PWI with phonological distractors (p. 12 line 159).

Our response:

As described in the introduction, changes of interference effects across repetitions have been reported for other paradigms, but not for PWI paradigms. The reported study, to our knowledge, is the only one addressing this in some formal way. We have rephrased this argument to make it clearer. 

Introduction, lines 194 ff: 

To our knowledge changes across repetitions have been formally addressed for a PWI paradigm only in one recent study (47). Using an auditory PWI design, interference effects are reported to be largely stable across naming repetitions with phonological distractors. This stands in contrast to the other paradigms mentioned above. Therefore we consider more evidence necessary to draw systematic conclusions about the stability of the interference effect in PWI paradigms.

 

Reviewer 2:

Overall impression:

Van Scherpenberg, Abdel Rahman and Obrig states that the main aim of their study was “to test whether semantic interference is confirmed for this novel paradigm, and whether the number of related words modulates its magnitude” (abstract); / “In the present study we focus on categorical semantic relations and investigate whether manipulating the extent of lexical activation within a lexical cohort modulates inhibition on subsequent picture naming.” (page 5). I wonder whether we really need new evidence showing that lexical access in language production is affected by semantic context and that this modulation can be incremental (e.g., Howard et al. 2006). Sincerely, I don’t think this aim would deserve publication in a journal as Plos ONE

In my opinion, the main interest of the current manuscript relies on the attempt to combine different naming paradigms to disentangle the nature of the semantic effect(s) in language production. To this respect, the topic discussed is timely and fits well with recent discussions in the field about the trade-off between accuracy and speed in picture naming (e.g., Nozari & Hepner, 2018). At the same time, the combined methodology of the study allows the comparison between word processing (eye tracking measure) and lexical access (naming latency measure) within the same task. In sum, the multi-novel experimental paradigm the authors adopt can be very interesting only if it is discussed in the context of current and relevant theoretical discussions.

Our response:

Thank you for this comment. Of course we agree that the mere reproduction of an effect well described in the literature was certainly not the aim of our current study. We believe this does become quite clear in the more detailed respective parts of the main manuscript but it indeed is not adequately introduced in the abstract. Therefore we have changed the respective paragraph in the abstract (line 30 ff.): 

To disentangle interacting effects of semantic context we combined different naming paradigms manipulating the number of competitors and assessing the effect of repeated naming instances. Evaluating processing of the cohort by eye-tracking provided us with a metric of the (implicit) recognition of the semantic cohort.

Comment 1:

The introduction is a little bit confusing and theoretical biased. As mentioned above, semantic effects (facilitation and interference) have been reported mainly in three experimental tasks (PWI, blocking/cyclic naming and continuous naming). In the first page of the Introduction (page 3), the authors focus on one model, the Swing Lexical Network (SLN), and introduce the semantic context effects in relation to this model. In particular, the SLN approach can explain facilitation and semantic effects in (all) experimental tasks depending on the trade-off between conceptual priming and lexical competition. Of course, I have no problems with this choice; but see Navarrete et al (2016) for a review of the same topic based on the distinction between inter-level trial semantic manipulation tasks (PWI) and intra-level semantic manipulation tasks (cyclic and continuous naming) without reference to any specific theoretical model.

Our response:

The design of our current study indeed relies on the three experimental paradigms and the theoretical framework supplied by the Swinging Lexical Network Model. Our paradigm is explicitly motivated by predictions from the SLN approach, namely on increased interference through additional members of a lexical cohort. We are not aware of studies combining features of the different paradigms in the way we do here. We believe it is helpful for the reader that the SLN is described in some detail, pointing out its parsimonious way to explain our findings. It is of course important to include alternative explanations of semantic interference effects and our present findings, and indeed we have included this information in various paragraphs in the introduction and discussion (for the respective changes please refer to the responses to Comment 1a below).

Comment 1a:

While the SLN assumes that the balance between two mechanisms (conceptual priming and lexical competition) is able to explain the diverse pattern of semantic effects, other proposals have made explicit the claim that different tasks entails different cognitive mechanisms to resolve lexical retrieval in semantic context. For instance, Mahon and colleagues (2007) have argued that semantic interference in the PWI can be explained by a post-lexical mechanism of self-monitoring (see also Dhooge & Hartsuiker, 2011); while semantic interference in picture naming tasks would arise at the semantic-to-lexical connections (through an inhibition mechanism that weakens these connections) (Navarrete et al. 2010; 2014). The authors need to mention this alternative(s) explanation(s).

Our response:

We agree that alternative models should be mentioned, although the SLN model seems most fit to accommodate all aspects of our findings. To highlight the alternative models we have now included a paragraph on these alternative explanations in the Introduction and referred back to them in the Discussion.

Introduction, lines 105ff:

Alternative explanations have claimed semantic facilitation to be the default effect, with semantic interference occurring only at post-lexical processing steps, where task-relevant (i.e. semantically related) responses to pictures have to be actively excluded (Dhooge & Hartsuiker, 2010; Finkbeiner & Caramazza, 2006; Mahon, Costa, Peterson, Vargas, & Caramazza, 2007). 

Discussion, lines 506ff:

Notably our findings also contrast with alternative explanations of the origins of semantic interference, specifically the response-exclusion hypothesis (Mahon et al., 2007). This theory posits that through frequent exposure, task-relevant responses (e.g. names of pictures from the same semantic category) need to be actively excluded from an articulatory output-buffer resulting in delayed naming. In our paradigm however, frequent exposure leads to faster naming, as we discuss in more detail below.

Comment 1b:

The experimental set entails the presentation of two events in this order: 1-the presentation of an array with several written words, and 2-the presentation of a picture target to be named. Previous literature has already explored to which extend written word processing influences the successive picture naming event. Again, further literature has to be included and mentioned:

Belke (2013); Navarrete et al. (2013); Vitkovitch et al., (2010); Vitkovitch & Cooper (2012)

Our response:

We thank the reviewer for pointing us to this important literature. We have included this literature in the introduction while also describing how our paradigm deviates from these studies. 

Introduction lines 121 ff:

The extent to which presenting a number of written words before naming pictures can influence picture naming speed has been investigated in previous research (Navarrete, Mahon, & Caramazza, 2010; Navarrete, Mahon, Lorenzoni, & Peressotti, 2016; Vitkovitch, Cooper-Pye, & Ali, 2010; Vitkovitch & Cooper, 2012). However, in these experiments words were presented consecutively and had to be overtly read out aloud. Moreover, the findings are partially contradictory. For example, Navarrete et al. (28 (Experiment 3), 29) found no transfer of interference from word to picture naming within one semantic category, whereas Vitkovitch et al. (Vitkovitch et al., 2010; Vitkovitch & Cooper, 2012) did report semantic interference for naming pictures after having named semantically related pictures. We here investigate how simultaneous presentation and lexical activation by reading (not producing) the words impact on the processing of the semantic relationships between the words and consecutive naming of un/related items.

Comment 1c:

(Page 6; lines 18-148). The interaction between semantic context and cycle observed in the cyclic naming tasks (facilitation first cycle, interference from the second cycle onward) has been extensively explored by Navarrete et al 2014. This has to be acknowledged.

Our response:

We regret not having cited the important discussion in the paper by Navarrete et al (2014). We indeed believe it is of relevance to our findings and have included the citation in the Introduction (line 166). Although we consider the relevance high for the general discussion the experiments in Navrarrete et al. focus on the blocked cyclic naming paradigm. To not overly complicate the paper, we do not discuss the findings in detail, since the discussion is not directly related to the predictions and findings from our own PWI-paradigm.

Comment 2a:

Page 5. Eye-mind hypothesis. The authors need to explain the “eye-mind hypothesis”. What does it means that participants should fixate on words longer which they have recognized to belong to the same category? Why? I would predict the reverse effect, less fixation time because of the semantic priming between related words.

Our response:

We agree with the reviewer that more information is needed regarding the eye-mind hypothesis. The point is also raised by reviewer 1 and we are grateful for this convergent suggestion which we believe a better description of our underlying rationale is absolutely relevant to correctly represent an overarching goal of the combined PWI and eye-tracking design. We have rephrased the respective paragraph (Introduction, lines 133 ff.) to (1) better illustrate the theoretical value of the eye-mind hypothesis for our study, and to (2) better explain why make hypotheses about different fixation durations depending on semantic relationship.

(1) The eye-mind hypothesis refers to the immediate relationship of fixation durations and processing intensity in reading and is the theoretical basis for our hypothesis that longer fixation durations to semantically related items should lead to stronger interference effects.

(2) Eye tracking has been used to assess semantic deficits in clinical populations, and studies have found that participants with semantic deficits fixated on semantically related and unrelated objects equally long. We hypothesize that our paradigm is suited to investigate this as well and that longer fixation durations to semantic category members compared to non-members indicate intact semantic processing abilities, i.e., semantic competence.

We hope that the paragraph addresses the reviewer’s concerns and makes our assumptions clearer. 

Introduction, lines 133 ff:

We proceed from the rationale that eye tracking can be used to investigate the semantic ‘competence’ of viewers. This assumption rests on paradigms performed in people with Primary Progressive Aphasia (PPA) and neurotypical controls. Suggesting impaired semantic memory abilities, participants suffering from PPA (Faria et al., 2018; Seckin et al., 2016) fixated on semantically unrelated objects (foils) more often and longer when compared to neurotypical controls, likely indicating difficulties to establish the semantic relationships between concepts. Here we hypothesize that the neurotypical young adults, are semantically competent, and should hence fixate on words longer which they have recognized to belong to the same category, when compared to the unrelated words. Thus analysis of fixation times was used to investigate the semantic ‘competence’ of the participants in each trial. Additionally we can use this measure to estimate the extent to which they activate the lexical cohort. According to the eye-mind hypothesis (Just & Carpenter, 1980; Rayner, 1995), readers’ gaze durations are immediately linked to what they are processing. That is, words that are fixated longer are also processed longer. We therefore predict that the longer participants fixate on related words belonging to the lexical cohort, the more activity will spread to this cohort and induce stronger competition resulting in inhibition on target selection.

Although the data are not yet fully analyzed, it may be of interest, that we have recently tested a cohort of people with a chronic acquired brain lesion in the left hemispheric extended language network with the paradigm introduced in the current paper. Preliminary analyses support our rationale to “operationalize” gaze preference as an indicator of lexico-semantic competence. What we find (with the caveat of preliminary analysis) is that this measure varies strongly across patients from no preference for related vs. unrelated items to patterns which are largely similar to our findings in the neurotypical group reported on here. 

Comment 2b:

Page 5. Eye-mind hypothesis. According to one of the references cited by the authors (Rayner, 1995), word frequency is one of the most critical variable determining the amount of time a word is fixed. Did the authors control the lexical frequency between the 8 words of the arrays? I guess that the amount of time I can fixate the words cat-dog-horse would depend on their own lexical frequency, as well as on the lexical frequency of the other words in the array. For instance, if in an array the non-related words are high frequency words (e.g. house-car-table-hand) I would have more/less time to fixate the related words (e.g., cat-dog-horse) than if the non-related words are low frequency words (e.g., skyscraper-tractor-stool-chin)

Our response:

The reviewer is right that word frequency is an important issue raised in Rayner (1995). An important difference with regard to our paper the respective paper mostly investigates sentence reading. Moreover, Just and Carpenter (1980) attribute the effect of word frequency mostly to lexical access and not (semantic) encoding. In the current study we did not specifically control for lexical frequency. However, all words have a very similar frequency (mean = 12.29, sd = 1.88, based on the Leipzig Corpora Collection (2011)). It should be highlighted though, that all stimuli were randomized across all conditions to avoid item-driven effects. They are also repeated several times as is explained in the manuscript. Moreover, items where included as random factors in the linear mixed model and participants were familiarized with all words and pictures before the experiment. We hope that this addresses the reviewer’s concerns.

Methods, lines 150 ff.:

To control for potential confounding effects all words used in the paradigm have a highly similar frequency: mean = 12.29, sd = 1.88, according to the Leipzig Corpora Collection (2011). Moreover, potential item-based effects are strongly attenuated by the fact that randomization was complete across conditions. 

Comment 2c:

Page 5. Eye-mind hypothesis. We are informed that: “We predict that the longer participants fixate on related words belonging to the lexical cohort, the more activity will spread to this cohort and induce stronger competition resulting in inhibition on target”. Justify this assumption

Our response:

We address this comment partly in our reply to comment 2a above. We assume that with more extensive processing of the distractor words, the connections between these concepts, compared to the unrelated concepts, should become stronger. A stronger, more closely connected lexical cohort will then result in increased competition.

Comment 2d:

The lack of interaction between Type array (related vs. unrelated) and Size (3, 4, 5 related words) seems problematic with the prediction that fixation times would increase according to the number of related words in the array. If I understood correctly: the authors predict that the average of fixation time for a related word (cat) should be longer when the word appears together with 4 related words (horse-pig-rabbit-dog) than when two related words (horse-pig). The problem is that the average-time a participant is looking at one word is not independent of the time the participant dedicates to look at the other words. Sorry if I miss something here!!!

Just to illustrate my point (arrays are presented for 6000 ms as in the experiment). If I fixate in a Size 3 – lexical cohort the word cat for 1000 ms, then the maximum fixation time I can dedicate to look at the other words of the array is 5000 ms; and for instance I look at the other two related words 2000 ms (horse 1000 ms and pig 1000ms). But if I fixate cat for 2000 ms in a Size 5 – lexica cohort, then the total time I can fixate the other words of the array is 4000 ms; and for instance I look at the other four related words 1000 ms (horse 250 ms, pig 250 ms, rabbit 250 ms, dog 250 ms). This hypothetical scenario serves to illustrate the point that, under the authors' assumption that fixation time depends on semantic context, the average fixation time for a single word cannot be independent of the semantic Size – lexical cohort! Therefore, it’s unclear what we can learn from the fixation measure. The authors need to clarify this confound.

Our response:

We thank the reviewer for illustrating this point. And indeed there may have been a confusion about our claims. We do not claim that the fixation time on each single related distractor word necessarily needs to be longer than each unrelated word, but that in total participants fixate longer on the lexico-semantic cohort compared to the other words. In other words: if there were no bias each word would be fixated for 1/8th of the total fixation time. What we show is that this is shifted for the related versus unrelated words of the set, with participants fixating longer on related words. The number of cohort members is cancelled out by this procedure. We have rephrased this description of our analysis in the Results section, hopefully making the analysis clearer.

Results, lines 416 ff:

For each trial, total viewing time recorded by the eye-tracker was ~4100 ms on average (i.e. ~1900ms participants did not fixate on any of the AoIs or data were not recorded). The measure can be assumed to depend on data quality (e.g. blinks) and attentional resources. Fig 4 shows the mean viewing times for each related and unrelated word across all trials and participants, and for each lexical cohort condition (3, 4 or 5 mutually related words out of 8 words in total in each trial). If there was no bias in fixating to members vs. non-members of the cohort, each word should be fixated for 1/8th of the total fixation time. The measures show that participants fixated longer on members than non-members, and therefore indicate the participant’s categorization skills of semantically related and unrelated words in each word set.

Reviewer 2 minor comments:

Statistics. Are the reported beta valuescorrect? Maybe they should be b values?!?

Our response:

In order to avoid confusion, the expression “beta” (β) has been changed to “estimate”. Thank you for pointing this potentially misleading wording!

 

References

Abdel Rahman, R., & Melinger, A. (2009). Semantic context effects in language production: A swinging lexical network proposal and a review. Language and Cognitive Processes, 24(5), 713–734. https://doi.org/10.1080/01690960802597250

Abdel Rahman, R., & Melinger, A. (2019). Semantic processing during language production: an update of the swinging lexical network. Language, Cognition and Neuroscience, 34(9), 1176–1192. https://doi.org/10.1080/23273798.2019.1599970

Leipzig Corpora Collection. (2011). German newspaper corpus based on material crawled in 2011. Leipzig Corpora Collection. Retrieved from https://corpora.uni-leipzig.de?corpusId=deu_newscrawl_2011

Dhooge, E., & Hartsuiker, R. J. (2010). The Distractor Frequency Effect in Picture-Word Interference: Evidence for Response Exclusion. Journal of Experimental Psychology: Learning Memory and Cognition, 36(4), 878–891. https://doi.org/10.1037/a0019128

Faria, A. V., Race, D., Kim, K., & Hillis, A. E. (2018). The eyes reveal uncertainty about object distinctions in semantic variant primary progressive aphasia. Cortex, 103, 372–381. https://doi.org/10.1016/j.cortex.2018.03.023

Finkbeiner, M., & Caramazza, A. (2006). Now you see it, now you don’t: On turning semantic interference into facilitation in a stroop-like task. Cortex, 42(6), 790–796. https://doi.org/10.1016/S0010-9452(08)70419-2

Just, M. A., & Carpenter, P. A. (1980). A theory of reading: From eye fixations to comprehension. Psychological Review. US: American Psychological Association. https://doi.org/10.1037/0033-295X.87.4.329

Mahon, B. Z., Costa, A., Peterson, R., Vargas, K. A., & Caramazza, A. (2007). Lexical Selection Is Not by Competition: A Reinterpretation of Semantic Interference and Facilitation Effects in the Picture-Word Interference Paradigm. Journal of Experimental Psychology: Learning Memory and Cognition, 33(3), 503–535. https://doi.org/10.1037/0278-7393.33.3.503

Navarrete, E., Mahon, B. Z., & Caramazza, A. (2010). The cumulative semantic cost does not reflect lexical selection by competition. Acta Psychologica, 134(3), 279–289. https://doi.org/10.1016/J.ACTPSY.2010.02.009

Navarrete, E., Mahon, B. Z., Lorenzoni, A., & Peressotti, F. (2016). What can Written-Words Tell us About Lexical Retrieval in Speech Production? Frontiers in Psychology, 6, 1982. https://doi.org/10.3389/fpsyg.2015.01982

Rabovsky, M., Schad, D. J., & Abdel Rahman, R. (2016). Language production is facilitated by semantic richness but inhibited by semantic density: Evidence from picture naming. Cognition, 146, 240–244. https://doi.org/10.1016/j.cognition.2015.09.016

Rayner, K. (1995). Eye movements and cognitive processes in reading, visual search, and scene perception. In Eye Movement Research (pp. 3–22).

Rose, S. B., Aristei, S., Melinger, A., & Abdel Rahman, R. (2018). The Closer They Are, the More They Interfere: Semantic Similarity of Word Distractors Increases Competition in Language Production. Journal of Experimental Psychology: Learning Memory and Cognition. https://doi.org/10.1037/xlm0000592

Seckin, M., Mesulam, M. M., Voss, J. L., Huang, W., Rogalski, E. J., & Hurley, R. S. (2016). Am I looking at a cat or a dog? Gaze in the semantic variant of primary progressive aphasia is subject to excessive taxonomic capture. Journal of Neurolinguistics, 37, 68–81. https://doi.org/10.1016/j.jneuroling.2015.09.003

Vitkovitch, M., Cooper-Pye, E., & Ali, L. (2010). The long and the short of it! Naming a set of prime words before a set of related picture targets at two different intertrial intervals. European Journal of Cognitive Psychology, 22(2), 161–171. https://doi.org/10.1080/09541440902743348

Vitkovitch, M., & Cooper, E. (2012). My Word! Interference from Reading Object Names Implies a Role for Competition during Picture Name Retrieval. Quarterly Journal of Experimental Psychology, 65(6), 1229–1240. https://doi.org/10.1080/17470218.2012.655699

---

## [Decision Letter · Decision Letter 1]

3 Feb 2020

PONE-D-19-20412R1

A novel multi-word paradigm for investigating semantic context effects in language production

PLOS ONE

Dear Ms van Scherpenberg,

Thank you for resubmitting your manuscript to PLOS ONE.  The resubmission was evaluated by the same two reviewers of the initial submission, and I have read it carefully myself.  The reviewers are split: Reviewer 2 feels that the submission is ready to be accepted, whereas Reviewer 1 still struggles to pinpoint the theoretically-motivated predictions that this work provides the appropriate way to address.  Although I am less expert in this area than both reviewers, I found the submission to be much improved, and the reviewers' criticisms to have been acted on in good faith.

I believe the work is near the point at which the field should decide on its merits.  That having been said, I am sympathetic to Reviewer 1's remaining concerns.  For instance, this reviewer points to the motivation given in the paragraph that starts on line 108: Perhaps this discussion could be wrapped up with an clearer indication of what types of theories would be ruled in or out in light of different possible experimental outcomes.  This would help readers understand why this particular task is the right one to pursue in light of the existing gaps in our understanding.  Similar concerns apply to the use of use of item repetition per the discussion around line 184: What is the space of possible findings, and how would they adjudicate between the different analyses on offer?  The reviewer offers several other constructive comments as well.  I have also included a few typos that I found while reading the manuscript in my comments below. 

In light of the reviews and my own thoughts as an ensemble, I'm taking the action of recommending Minor Revision for the manuscript, to give you the opportunity to further address Reviewer 1's worries to the extent you find appropriate.  In taking this action, I do not intend to send a revised manuscript out for further external review, and am optimistic about the eventual outcome.

We would appreciate receiving your revised manuscript by Mar 19 2020 11:59PM. To enhance the reproducibility of your results, we recommend that if applicable you deposit your laboratory protocols in protocols.io, where a protocol can be assigned its own identifier (DOI) such that it can be cited independently in the future. For instructions see: http://journals.plos.org/plosone/s/submission-guidelines#loc-laboratory-protocols

We look forward to receiving your revised manuscript.

Kind regards,

Andrew Kehler, Ph.D

Academic Editor

PLOS ONE

Additional Editor Comments (if provided):

Here are a few typos I caught while reading the manuscript:

line 162: delete extra period

line 407: it looks like a caption got cut-and-pasted here

line 429: "fixations durations"

line 457: underline extends to subsequent space

line 458: adjoin "competitors" to "members/" 

line 466: remove "(ii)"

Reviewers' comments:

Reviewer's Responses to Questions

**Comments to the Author**

1. If the authors have adequately addressed your comments raised in a previous round of review and you feel that this manuscript is now acceptable for publication, you may indicate that here to bypass the “Comments to the Author” section, enter your conflict of interest statement in the “Confidential to Editor” section, and submit your "Accept" recommendation.

Reviewer #1: (No Response)

Reviewer #2: All comments have been addressed

2. Is the manuscript technically sound, and do the data support the conclusions?

Reviewer #1: No

Reviewer #2: Yes

3. Has the statistical analysis been performed appropriately and rigorously? 

Reviewer #1: Yes

Reviewer #2: Yes

4. Have the authors made all data underlying the findings in their manuscript fully available?

Reviewer #1: Yes

Reviewer #2: No

5. Is the manuscript presented in an intelligible fashion and written in standard English?

Reviewer #1: Yes

Reviewer #2: Yes

6. Review Comments to the Author

Reviewer #1: Overall, I found the revised manuscript still lacking in terms of clear predictions that are theoretically motivated, and conclusions that are well-supported by the data. Below I explain at what points I did not follow the rationale.

Theoretical importance of the novel aspects of the paradigm. The first paragraph of the introduction does a good job in explaining how the novel paradigm is novel in comparison to previous work. However, it is vague in explaining the theoretical importance for the novelty, that is, why is it important to, for example “explore this finding [semantic context effects in naming] further”; or as it put in the general discussion, why it is important “to investigate whether and how semantic interference effects through categorical distractors can be modulated”. What question(s) are left open by previous research that this novel paradigm will now be able to address (e.g., how does the current study address a gap (s) in previous studies)? Below I provide examples where the theoretical rationale and importance for the novel aspects of the paradigm (increasing number of distractors, repetition of pictures, eye-gaze fixation measures) continue to be unclear throughout the manuscript.

1) Increasing numbers of distractors

For example in the introduction (p.19 line 108), the authors provide a vague description of the theoretical importance of “manipulating the extent of lexical activation within a lexical cohort”. Please address if successful (in manipulating the extent of lexical activation within a lexical cohort via increasing numbers of word distractors), what this outcome will support in terms of our understanding of word production dynamics and critically, what the result will refute.

The ms (p. 19, line 120) mentions that previous work increased distractors in a PWI set up but that this manipulation is [presumably] irrelevant because the words had to be read out loud, not just passively read. Please describe why reading out loud vs. passively is an important manipulation in terms of testing different predictions concerning how word production (semantic interference) proceeds at the semantic and lexical levels.

2) Item repetition

Please provide a more specific rationale to explain the necessity of including repetition in a PWI paradigm (p. 22, line 184): “ we consider more evidence necessary to draw systematic conclusion about the stability of the interference effect in PWI paradigms”. In the introduction as written, it is still unclear how finding an effect of repetition fits or does not fit with theoretical predictions which provide evidence for or against a hypothesis. For example, it could be clearly stated that “if we find that interference increases across repetitions this will support an X interpretation of lexical selection/interference but will not support a Y interpretation. Alternatively, if we find that interference decreases across repetitions this supports a Y interpretation but not X interpretation”. The response exclusion hypothesis is briefly discussed in the general discussion (p. 37, line 514) but two sentences was not enough for me to follow the argument.

The motivation for the repetition factor in this new PWI paradigm is different in the introduction vs. the general discussion. In the general discussion, the authors argue that because repetition affects semantic interference in blocked cyclic and continuous naming, it should be explored in this PWI paradigm variant. However, the repeated exposure in the continuous/blocked cyclic naming paradigm is one of exposure to the same semantic category during naming, not necessarily the same items being named (cf. blocked cyclic paradigm). Thus, I do not clearly follow the theoretical comparison between repeated naming of the same pictures in a PWI paradigm format vs. the repeated retrieval from the same semantic category of different items in continuous/blocked cyclic naming paradigms.

The authors explain that semantic interference disappears after an item is first named because facilitation (during the SR trials) increasingly neutralizes the interference effect (during the SR trials) across repetitions. However, under this interpretation, do the authors predict that facilitation should eventually “win” against interference with multiple repetitions and make the SR condition faster than the UR condition as repetitions increase? The results demonstrate that after the first repetition the SR/UR trials RTs are virtually identical. It is not clear what the predictions were and what the results support.

3) Eye gaze

Word distractor eye-fixations are introduced in the paradigm to measure “semantic competence”. Semantic competence is referred to as semantic processing abilities or the ability “to establish the semantic relationships between concepts”. However, I still find the rationale confusing. This is because the rationale here is that in neurotypical populations “longer fixation durations…indicate intact semantic processing abilities” but longer fixation durations in clinical populations indicate difficulties to “establish the semantic relationships between concepts”. This logic is the opposite from that used to describe how neurotypical and brain-damaged subjects perform in blocked-cyclic naming where the pattern is assumed to be the same (i.e., longer RTs and/or more errors in the semantically related vs. unrelated blocks for both populations are reflective of the same underlying mechanisms, i.e. semantic interference during naming). Thus, it is unclear how the eye gaze = increased connection strength at the semantic level = better semantic competence argument works.

Regarding the finding that subjects spent more time looking at words that were semantically related vs. unrelated, the authors conclude that this resolves a debate into the literature as to whether Ss implicitly categorize words when not specifically instructed. However, there is no mention in the manuscript concerning the previous debate concerning this point, how the current results rectify this gap in the literature, nor a clear explanation of how implicit “word categorization” occurs with respect to what is theorized to occur in the PWI paradigm.

Minor point:

Because 1) the design included a repetition component in order to increase activation, 2) the authors report and interpret the interaction between semantic relatedness (SR/UR) and naming repetitions (times points 2-6), and 3) the general discussion refers to the change (or lack thereof) in the SR/UR conditions across repetitions, please include (supplemental or otherwise) the figure including both SR and UR conditions across all repetitions.

Reviewer #2: I have read the new version of the manuscript. The authors have addressed the points that were raised.

7. PLOS authors have the option to publish the peer review history of their article (what does this mean?). If published, this will include your full peer review and any attached files.

Reviewer #1: No

Reviewer #2: No

---

## [Author Response · Author response to Decision Letter 1]

26 Feb 2020

Reviewer 1

Overall impression: 

Overall, I found the revised manuscript still lacking in terms of clear predictions that are theoretically motivated, and conclusions that are well-supported by the data. Below I explain at what points I did not follow the rationale.

Theoretical importance of the novel aspects of the paradigm. The first paragraph of the introduction does a good job in explaining how the novel paradigm is novel in comparison to previous work. However, it is vague in explaining the theoretical importance for the novelty, that is, why is it important to, for example “explore this finding [semantic context effects in naming] further”; or as it put in the general discussion, why it is important “to investigate whether and how semantic interference effects through categorical distractors can be modulated”. What question(s) are left open by previous research that this novel paradigm will now be able to address (e.g., how does the current study address a gap (s) in previous studies)? Below I provide examples where the theoretical rationale and importance for the novel aspects of the paradigm (increasing number of distractors, repetition of pictures, eye-gaze fixation measures) continue to be unclear throughout the manuscript.

Our response:

We thank the reviewer for the feedback and hope that our contributions and answers to the comments below address the reviewer’s concerns. Especially, we have elaborated on the novelty of changing only the number of distractors, but keeping the activation strength stable, to investigate mechanisms of lexical competition. Moreover, we have clarified once more our predictions regarding the eye tracking measures and refer to our exploratory analysis of naming repetition in our paradigm.

Comment1: 

1) Increasing numbers of distractors

For example in the introduction (p.19 line 108), the authors provide a vague description of the theoretical importance of “manipulating the extent of lexical activation within a lexical cohort”. Please address if successful (in manipulating the extent of lexical activation within a lexical cohort via increasing numbers of word distractors), what this outcome will support in terms of our understanding of word production dynamics and critically, what the result will refute.

Our response:

According to the Swinging Lexical Network model (Abdel Rahman & Melinger, 2009, 2019), the amount of semantic interference through lexical competition is dependent on (1) the activation strength of lexical competitors and (2) the number of competing items. By keeping the activation strength stable (by using semantically closely related stimuli) in this study we are able to tackle the question whether the number of lexical competitors has a direct influence on the amount of semantic interference. This paragraph is now added to the introduction:

Introduction, lines 110ff:

Using a set of closely related entries of a number of lexical cohorts the activation strength per item can be assumed largely homogeneous. Using these sets we parametrically change the number of distractors to investigate, whether this has a direct influence on the amount of semantic interference.

Comment 2:

The ms (p. 19, line 120) mentions that previous work increased distractors in a PWI set up but that this manipulation is [presumably] irrelevant because the words had to be read out loud, not just passively read. Please describe why reading out loud vs. passively is an important manipulation in terms of testing different predictions concerning how word production (semantic interference) proceeds at the semantic and lexical levels.

Our response:

Our main argument for these other studies to differ importantly from our design is the fact that the word stimuli were presented consecutively instead of simultaneously. We agree that reading aloud vs quietly may not be the key difference in modulation between previous and our designs, although differences have been described (Laubrock & Kliegl, 2015). Reading words out loud may activate the full preparatory and motor act of production similar to picture naming, therefore encompassing other effects like repetition priming. 

Comment 3: 

2) Item repetition

Please provide a more specific rationale to explain the necessity of including repetition in a PWI paradigm (p. 22, line 184): “ we consider more evidence necessary to draw systematic conclusion about the stability of the interference effect in PWI paradigms”. In the introduction as written, it is still unclear how finding an effect of repetition fits or does not fit with theoretical predictions which provide evidence for or against a hypothesis. For example, it could be clearly stated that “if we find that interference increases across repetitions this will support an X interpretation of lexical selection/interference but will not support a Y interpretation. Alternatively, if we find that interference decreases across repetitions this supports a Y interpretation but not X interpretation”. The response exclusion hypothesis is briefly discussed in the general discussion (p. 37, line 514) but two sentences was not enough for me to follow the argument.

Our response:

We have rephrased the quoted paragraph as follows:

Introduction, lines 183ff:

In contrast to these paradigms, to our knowledge, for a PWI paradigm changes across repetitions have been formally addressed only in one recent study (Kurtz, Schriefers, Mädebach, & Jescheniak, 2018). Using an auditory PWI design, interference effects are reported to be largely stable across naming repetitions of the same pictures with phonological distractors. This stands in contrast to the other paradigms mentioned above, and systematic conclusions about the stability of the interference effect in PWI paradigms can only be tentative at present.

Considering the scope of this paper and the lack of previous findings on repetition in PWI, we don’t feel comfortable formulating clear hypotheses about the meaning of a repetition effect for word production models. However, we have elaborated on the discussion of the response-exclusion hypothesis and our opinion of why it is unfitting to our findings:

Discussion, lines 515ff:

Notably our findings also contrast with alternative explanations of the origins of semantic interference, specifically the response-exclusion hypothesis (Mahon, Costa, Peterson, Vargas, & Caramazza, 2007). This theory posits that through frequent exposure, task-relevant responses (e.g. names of pictures from the same semantic category) need to be actively excluded from an articulatory output-buffer resulting in delayed naming. In our paradigm, these task-relevant items would include previously named pictures, and previously fixated words that were part of a category word set. However, as we discuss in more detail below, our results show that indeed frequent exposure to the material leads to faster naming thus making an explanation of an effortful and therefore inhibitory monitoring mechanism unlikely. .

Comment 4: 

The motivation for the repetition factor in this new PWI paradigm is different in the introduction vs. the general discussion. In the general discussion, the authors argue that because repetition affects semantic interference in blocked cyclic and continuous naming, it should be explored in this PWI paradigm variant. However, the repeated exposure in the continuous/blocked cyclic naming paradigm is one of exposure to the same semantic category during naming, not necessarily the same items being named (cf. blocked cyclic paradigm). Thus, I do not clearly follow the theoretical comparison between repeated naming of the same pictures in a PWI paradigm format vs. the repeated retrieval from the same semantic category of different items in continuous/blocked cyclic naming paradigms.

Our response:

In order to avoid a potential misinterpretation of our claims that the repetition effect in PWI and other paradigms might stem from the same origin, we have added a sentence to the discussion for clarification: 

Note, however, that this sort of cumulative interference results from the repetition of categories, not single items. (Discussion, lines 515f.) 

The design of our paradigm and many other PWI paradigms requires the repetition of items due to randomization. While researching potential consequences of this design feature, we failed to find reports in the literature. We therefore recommend this to be further addressed by future studies. 

Comment 5:

The authors explain that semantic interference disappears after an item is first named because facilitation (during the SR trials) increasingly neutralizes the interference effect (during the SR trials) across repetitions. However, under this interpretation, do the authors predict that facilitation should eventually “win” against interference with multiple repetitions and make the SR condition faster than the UR condition as repetitions increase? The results demonstrate that after the first repetition the SR/UR trials RTs are virtually identical. It is not clear what the predictions were and what the results support.

Our response:

Due to the lack of previous findings on repetition effects in PWI, and the fact that our paradigm is a novel variation of the classical paradigm, we see the inclusion of the factor “repetition” more as an exploratory analysis, and would like to remain with the predictions that are already stated in the Introduction (lines 203ff.)

Comment 6: 

3) Eye gaze

Word distractor eye-fixations are introduced in the paradigm to measure “semantic competence”. Semantic competence is referred to as semantic processing abilities or the ability “to establish the semantic relationships between concepts”. However, I still find the rationale confusing. This is because the rationale here is that in neurotypical populations “longer fixation durations…indicate intact semantic processing abilities” but longer fixation durations in clinical populations indicate difficulties to “establish the semantic relationships between concepts”. This logic is the opposite from that used to describe how neurotypical and brain-damaged subjects perform in blocked-cyclic naming where the pattern is assumed to be the same (i.e., longer RTs and/or more errors in the semantically related vs. unrelated blocks for both populations are reflective of the same underlying mechanisms, i.e. semantic interference during naming). Thus, it is unclear how the eye gaze = increased connection strength at the semantic level = better semantic competence argument works.

Our response:

We apologize if our hypotheses regarding the eye tracking measures were still unclear. However, as we describe in the introduction, our assumption about fixation durations to semantically related vs unrelated items reflecting semantic competence is informed by studies on clinical populations (Faria, Race, Kim, & Hillis, 2018; Seckin et al., 2016). These studies showed that impaired semantic knowledge leads to blurry distinctions between semantic categories reflected in longer fixations to foils, i.e. more equal fixation patterns for related and unrelated items. We simply posit that neurotypical young adults have intact semantic knowledge and should therefore should a clear distinct pattern between fixations to related vs. unrelated words. This hypothesis is confirmed by our data. 

Comment 7: 

Regarding the finding that subjects spent more time looking at words that were semantically related vs. unrelated, the authors conclude that this resolves a debate into the literature as to whether Ss implicitly categorize words when not specifically instructed. However, there is no mention in the manuscript concerning the previous debate concerning this point, how the current results rectify this gap in the literature, nor a clear explanation of how implicit “word categorization” occurs with respect to what is theorized to occur in the PWI paradigm.

Our response:

We may have missed the point and unfortunately fail to find the specific paragraph the reviewer refers to. Probably we should point out that our claims regarding the eye-tracking findings are quite modest and we certainly do not claim to resolve a debate in the literature. Using eye-tracking we simply aimed to assess a measure of how participants process the set of words presented. From our review of literature on participants with impaired language and semantic abilities we concluded that the implicit recognition of a cohort can be deduced from a longer fixation on related compared to unrelated words. Our question was on how the intensity of processing is also augmented by longer fixation (likely indicating a more intense processing). Here our finding is that implicit categorization seems to have no further effect on semantic interference (i.e. there is no correlation between fixation durations on semantically related words and reaction times). 

Comment 8:

Minor point:

Because 1) the design included a repetition component in order to increase activation, 2) the authors report and interpret the interaction between semantic relatedness (SR/UR) and naming repetitions (times points 2-6), and 3) the general discussion refers to the change (or lack thereof) in the SR/UR conditions across repetitions, please include (supplemental or otherwise) the figure including both SR and UR conditions across all repetitions.

Our response:

The figure below is now included in the appendix of the paper. 

Literature

Abdel Rahman, R., & Melinger, A. (2009). Semantic context effects in language production: A swinging lexical network proposal and a review. Language and Cognitive Processes, 24(5), 713–734. https://doi.org/10.1080/01690960802597250

Abdel Rahman, R., & Melinger, A. (2019). Semantic processing during language production: an update of the swinging lexical network. Language, Cognition and Neuroscience, 34(9), 1176–1192. https://doi.org/10.1080/23273798.2019.1599970

Faria, A. V., Race, D., Kim, K., & Hillis, A. E. (2018). The eyes reveal uncertainty about object distinctions in semantic variant primary progressive aphasia. Cortex, 103, 372–381. https://doi.org/10.1016/j.cortex.2018.03.023

Kurtz, F., Schriefers, H., Mädebach, A., & Jescheniak, J. D. (2018). Incremental learning in word production: Tracing the fate of non-selected alternative picture names. Journal of Experimental Psychology: Human Perception and Performance, 44(10), 1586–1602. https://doi.org/10.1037/xhp0000558

Laubrock, J., & Kliegl, R. (2015). The eye-voice span during reading aloud. Frontiers in Psychology, 6, 1432. https://doi.org/10.3389/fpsyg.2015.01432

Mahon, B. Z., Costa, A., Peterson, R., Vargas, K. A., & Caramazza, A. (2007). Lexical Selection Is Not by Competition: A Reinterpretation of Semantic Interference and Facilitation Effects in the Picture-Word Interference Paradigm. Journal of Experimental Psychology: Learning Memory and Cognition, 33(3), 503–535. https://doi.org/10.1037/0278-7393.33.3.503

Seckin, M., Mesulam, M. M., Voss, J. L., Huang, W., Rogalski, E. J., & Hurley, R. S. (2016). Am I looking at a cat or a dog? Gaze in the semantic variant of primary progressive aphasia is subject to excessive taxonomic capture. Journal of Neurolinguistics, 37, 68–81. https://doi.org/10.1016/j.jneuroling.2015.09.003

---

## [Editor Report · Decision Letter 2]

2 Mar 2020

A novel multi-word paradigm for investigating semantic context effects in language production

PONE-D-19-20412R2

Dear Dr. van Scherpenberg,

We are pleased to inform you that your manuscript has been judged scientifically suitable for publication and will be formally accepted for publication once it complies with all outstanding technical requirements.

With kind regards,

Andrew Kehler, Ph.D

Academic Editor

PLOS ONE

---

## [Editor Report · Acceptance letter]

25 Mar 2020

PONE-D-19-20412R2 

A novel multi-word paradigm for investigating semantic context effects in language production 

Dear Dr. van Scherpenberg:

I am pleased to inform you that your manuscript has been deemed suitable for publication in PLOS ONE. Congratulations! Your manuscript is now with our production department. 

With kind regards,

on behalf of

Dr. Andrew Kehler 

Academic Editor

PLOS ONE